# Garden-path sentences and the diversity of their (mis)representations

**Markéta Ceháková**[1,2]*, **Jan Chromý**[1,2]*

**1** Institute of Czech Language and Theory of Communication, Faculty of Arts, Charles University, Prague, Czech Republic, **2** ERCEL Lab, Faculty of Arts, Charles University, Prague, Czech Republic

☉ These authors contributed equally to this work.
* marketa.cehakova@ff.cuni.cz (MC); jan.chromy@ff.cuni.cz (JC)

**Data Availability Statement:** Experimental items, data and R scripts used for their analysis are accessible in the following repository: https://osf. io/98g26/ [DOI: 10.17605/OSF.IO/98G26].

**Funding:** JC was supported by the Alexander von Humboldt Foundation. The funding was also

## Abstract

Previous studies have reliably shown that the initial misanalysis of garden-path sentences lingers even after the whole sentence is processed. However, other aspects of the resulting representation of these sentences are far from being clear. Some authors argue that comprehenders form a full analysis of the sentence which is faithful to the input and that the fact that the misanalysis lingers is due to an inhibition failure. Recently, it has been shown that comprehenders might not manage to create a coherent representation at all, at least in the case of more demanding garden-path structures. The aim of the current paper is to examine resulting representations of garden-path sentences in more detail. To do this, four self-paced reading experiments in Czech were conducted, which differed in the presentation mode (word-by-word and sentence-at-once) and comprehension question format (yes–no questions and open-ended questions). The experiments replicated effects typical for the lingering initial misanalysis, but provided mixed evidence for other aspects of resulting representations. In most cases, participants managed to build a coherent representation that was faithful to the input. However, both the quantitative and qualitative analysis of the results showed that comprehenders sometimes maintained multiple local interpretations at once or even failed to build a coherent representation of a garden-path sentence. Thus, we argue that resulting representations of garden-path sentences are in fact not uniform, but rather diverse, and they vary both in their faithfulness to the presented input and in their internal coherence.

## Introduction

Garden-path sentences like (1) have been the focus of psycholinguistic studies for many years thanks to their specific structure containing a local syntactic (or semantic) ambiguity that initially leads comprehenders to interpret the sentence in a way that is not consistent with the following input (e.g., 'Anna dressed the baby'). This misanalysis only tends to be resolved and reinterpreted later in the sentence. The temporary nature of the ambiguity allows researchers to probe various aspects of sentence processing, including the extent to which comprehenders are able to form coherent interpretations faithful to the input after realizing their initial analysis was wrong.

provided by the Charles University institutional program Cooperatio The funders had no role in study design, data collection and analysis, decision to publish, or preparation of the manuscript.

**Competing interests:** The authors have declared that no competing interests exist.

(1) While Anna dressed the baby played in the crib.

The initial misinterpretation of the sentence (1) is caused by the ambiguity of the verb *dressed*, which can be temporarily interpreted either as transitive, with *the baby* as a patient, or as reflexive, with *Anna* being both the agent and the patient. Because the transitive interpretation is preferred, comprehenders initially analyze *the baby* as the direct object of the verb *dressed*. This can be influenced by several factors, including syntactic complexity [1, 2], the frequency of the given verb form [3, 4], or plausibility of *the baby* as an argument of the verb [5, 6]. However, this initial analysis turns out to be incorrect, because the verb *played* would lack a subject if it was maintained.

According to Fodor and Inoue [7], the parser uses an operation called *Attach anyway* to attach the verb *played* to the current structure once it is encountered even though the result is ungrammatical. Then, it tries to repair the ungrammaticality by finding a suitable subject, in this case *the baby*. *The baby* is detached from its initial object position and attached again as the subject of *played*. Because the transitive verb *dressed* now lacks a direct object, a new ungrammaticality is created, which is solved by finding a reflexive interpretation of the verb. In general, the parser can continue adjusting the structure until the result is grammatical or until it encounters an ungrammaticality that cannot be repaired, in which case the reanalysis fails and the whole sentence is considered ungrammatical. Some approaches, e.g. cue-based retrieval, also argue that the difficulty and success of reanalysis can depend on the amount of interference caused by elements that are syntactically and semantically similar to the one that needs to be retrieved by the verb *played* [8–10].

It had long been assumed that comprehenders either manage to arrive at a full and faithful interpretation after they perform reanalysis, or they fail to build a coherent representation of the sentence altogether. However, as Christianson et al. [11] showed in their study, this might not be the case. They argue that reanalysis can be carried out only partially in some cases, resulting in interpretations that are not perfect but only "good enough" for the given situation. For example, for sentences like (1), the reanalysis might stop after finding a suitable subject for the verb *played*, but the initial interpretation of *the baby* as an object of *dressed* might not be erased, i.e., the initial misinterpretation of the garden path might still linger even after the whole sentence was processed.

These results were replicated in a study by Christianson et al. [12] which focused mostly on differences in processing between younger and older speakers. The tendency of the initial misanalysis to linger was also shown in other studies with various experimental designs such as paraphrasing [13], syntactic priming [14], multiple-choice questions [15], or the picture matching task [6]. A similar effect was also shown for L2 speakers of English [16–19]; and also for L1 speakers of other languages [15, 20].

Slattery et al. [21] suggest a different explanation for the lingering misanalysis. Unlike Christianson et al. [11], who propose that the syntactic structure built during reanalysis might be inaccurate or incomplete, the authors of this study argue that reanalysis does, in fact, lead to the creation of a syntactic structure that is complete, globally coherent and faithful to the input. According to them, reanalysis is incomplete in a different aspect, namely in that comprehenders fail to delete a memory representation built on the basis of the initial misanalysis, and this memory trace later interferes with the new representation of the whole sentence. The authors used the Gender Mismatch Effect to test this hypothesis in an eye-tracking experiment. In sentences like (2), a full and faithful syntactic structure needs to be built for the parser to detect a gender mismatch between the reflexive pronoun (*himself*) and its antecedent (*David's mother*); specifically, *David's mother* needs to be interpreted as the subject of the matrix clause.

(2) After the bank manager telephoned David's mother grew worried and gave himself approximately five days to reply.

The experiment has revealed that the Gender Mismatch Effect (in the form of longer first-pass reading times on the reflexive) is present in both the ambiguous and the unambiguous condition. The authors interpret this finding as an evidence that a full and faithful syntactic representation of the garden-path sentence was built. They also argue that the initial misanalysis remains active even after the reanalysis was performed, and that it interferes with new material. This can be seen in sentences like (3), where comprehenders usually slow down on the region *himself*.

(3) While Frank dried off the truck that was dark green was peed on by a stray dog. Frank quickly finished drying himself off then yelled out the window at the dog.

The authors explain this slow-down as an effect of interference caused by the initial misanalysis, which contains information that is contradictory to the information presented in later parts of the text. A similar effect was detected by Sturt [22].

It has been pointed out [20, 23] that Slattery et al. [21] did not use questions to test comprehension and that thus they could not be sure whether the sentences were really misinterpreted. The Gender Mismatch Effect thus could have been caused only by those trials in which the sentences were analyzed faithfully to the input.

The problem of the absence of comprehension questions was addressed by Qian et al. [24], who conducted a conceptual replication of the study by Slattery et al. and arrived at similar conclusions after examining the form of the final interpretation by combining self-paced reading and ERP measuring experiments with comprehension questions.

Another replication was reported by Huang and Ferreira [23], who carried out two experiments that combined online processing measures (eye-tracking, self-paced reading) with comprehension questions targeting the initial misanalysis, using the same stimuli as Slattery et al. [21]. Incorrectly answered trials were analyzed separately from those that were answered correctly. This allowed the authors to link the presence of the Gender Mismatch Effect (which indicates that a proper syntactic structure was built) with the response accuracy (which indicates whether the sentence was interpreted in accordance with the input). The Gender Mismatch Effect was present in both correctly and incorrectly answered trials, and its presence was independent of the grammatical features of other nouns in the sentence, suggesting that the parser does not ignore syntactic constraints while searching for an antecedent. These results are in accordance with the hypothesis of Slattery et al. [21].

Huang and Ferreira [23], however, point out that there is in fact no need for a full syntactic representation of the sentence to be built for the Gender Mismatch Effect to appear. As has been said above, it is crucial that the ambiguous noun (e.g. *David's mother*) is analyzed as the subject of the matrix clause for it to serve as the antecedent for the reflexive. It does not, by itself, mean that the previous analysis (with the noun as the object) has to be deleted and that the whole sentence has to be reanalyzed faithfully. The Gender Mismatch Effect would still be present if the reanalysis was not performed until the end and instead stopped after a suitable noun for the subject position was identified. It is thus possible that the parser only builds several locally coherent structures (e.g., *the bank manager telephoned David's mother + David's mother grew worried*). These structures cannot be combined into a single global one without violating the rules of grammar, but each of them can generate its own interpretation for the given part of the sentence. Current data are in accordance with the explanations given by both Slattery et al. [21] and Huang and Ferreira [23].

It should be noted that most of the research on garden-path sentences within the good-enough approach is focused on studying a specific type of local ambiguity: one where the combination of a verb and a noun is initially interpreted as a transitive verb and its direct object, and later has to be reinterpreted as an intransitive or reflexive verb and the subject of the matrix clause, or, in the case of Sturt's experiment [22], as a transitive verb with a subordinate clause in the position of the direct object.

A different type of garden-path structures has been studied by Christianson and Luke [25], who worked with sentences containing an ambiguous noun phrase coordination like (4).

(4) The publisher called up the editor and the author refused to change the book's ending.

In sentence (4) the whole phrase *the editor and the author* is interpreted as the direct object of the verb *called up*. However, as it later turns out, the conjunction *and* does not connect two noun phrases but two clauses. The participants were once again asked to answer a comprehension question targeting the initial misanalysis (*Did the publisher call the editor and the author?*). The response accuracy was much higher in this case, around 80%, suggesting that the tendency of the initial misanalysis to linger was less strong than for the structure used in the studies mentioned above, where the response accuracy reached only 50%.

An even lower response accuracy (around 20%) was produced by the participants in a study by Christianson et al. [26]. This study used sentences like (5) with yet another type of ambiguity where a reduced relative clause *tossed the ball* is at first interpreted as a continuation of the matrix clause (predicate and direct object).

(5) The player tossed the ball interfered with the other team.

Another type of temporarily ambiguous structure was examined by Fujita and Cunnings [19]. The authors examined whether the initial misanalysis persists in the case of filled-gap and non-filled-gap sentences, such as (6) and (7), when compared to their unambiguous counterparts, such as (8).

(6) Elisa noticed the truck which the policeman watched the car from earlier that morning.

(7) Elisa noticed the truck which the policeman watched very quietly from earlier that morning.

(8) Elisa noticed the truck from which the policeman watched the car earlier that morning.

Sentences like (6) and (7) contain long-distance dependencies where an argument, in this case the filler 'the truck', is dislocated from a post-verb region of a relative clause, the gap (which is located after the preposition *from* in our case). In order to interpret the sentences correctly, comprehenders need to link the filler and the gap. Since comprehenders tend to assign a filler to a gap as soon as possible, they usually tend to assume the policeman watched the truck. However, this representation is proven incorrect by the presence of the preposition, and in case of sentence (6) also by the presence of *the car*.

In a series of experiments working both with online and offline comprehension measures, the authors show that the initial misanalysis has a tendency to linger for this type of sentence. When asked about the initial misanalysis, *e.g. Did the policeman watch the truck?*, the participants tended to answer correctly in approximately 70% of the cases with filled-gap sentences like (6). In cases with non-filled-gap sentences, the response accuracy varied between 30% and 70% depending on the experimental design.

The results of these studies suggest that garden-path structures may vary in difficulty and by extension in the form of the final representation. For some structures, it even seems that a

coherent syntactic representation is not formed at all. For example, in the case of certain rather demanding Czech garden-path structures, comprehenders might even give up on trying to form a coherent representation of any kind and instead rely on general inference when making sense of the sentence [20].

This difference can be caused by typological characteristics of the two languages (unlike English, Czech is a language with a relatively free word order and a rich morphology), but it can also be linked to linguistic properties of the given garden-path structures. As Chromý [20] argues, resulting representations of garden-path sentences may vary based on the difficulty of the structure. It may be the case that for some structures, a coherent representation is not formed at all; for some, reanalysis results in a full and coherent syntactic representation that is faithful to the input, but the initial misinterpretation lingers; for others, both the syntactic representation and interpretation may be successfully repaired.

There is one more issue that we would like to point out. The previous studies have typically approached the question of lingering misanalysis by trying to find a single explanation for the effects of garden-path sentences on answering comprehension questions. At least three possibilities were introduced: (1) Reanalysis leads to the creation of several partial syntactic structures that are locally coherent but that cannot be combined into a single grammatical structure. These partial structures generate interpretations for given parts of the sentence. (2) Reanalysis leads to the creation of a globally coherent syntactic structure that is grammatically correct and faithful to the input. At the same time, comprehenders fail to erase the interpretation built during the initial misanalysis. (3) Readers fail to generate a coherent representation of the sentence (at least in the case of more complicated garden-path sentences) and rely on the lexical content of the sentence retained in their memory, general inference and encyclopedic knowledge when answering the questions. The previous studies do not explicitly claim that the resulting representation always looks the same. For example, Christianson et al. [12] or Malyutina and den Ouden [6] explicitly examine individual differences between comprehenders. Nevertheless, the authors still focus on trying to find a prototypical, general, model explanation for what happens when people are garden-pathed (for example the explanations offered by Christianson et al. [11] and Slattery et al. [21] are tested against each other, suggesting they are mutually exclusive).

However, there might be an alternative approach to the issue at hand. Perhaps, a single explanation is too simplistic. The reasons why readers answer comprehension questions incorrectly may differ from participant to participant and also from item to item. Since the current data cannot reliably distinguish between such reasons, it is possible that all of the potential explanations mentioned above are in fact not mutually exclusive. They all can represent different processing outcomes that readers arrive to after encountering garden-path sentences. This cannot be addressed properly if we analyze only the grouped data and if we use solely yes–no questions, because these do not tell us much about the resulting representation. Our approach will overcome this by focusing on the use of a combination of quantitative and qualitative analyses, namely open-ended comprehension questions, to examine the resulting representations of garden-path sentences in their full variety, without limiting the comprehenders' options by asking them whether they agree or disagree with a specific analysis of a given sentence. This would allow us to treat the resulting representations of garden-path sentences like a scale with a variety of options, ranging from full and faithful representations to incoherent ones, instead of focusing on finding a prototypical explanation for the phenomenon. We could thus see whether a single explanation may account for all (or most of) the incorrect answers that seem to suggest the presence of lingering misanalysis or whether the types of incorrect answers are rather diverse and cannot be explained unitarily.

In the current study, we thus introduce a set of experiments that focus on probing the final representation of Czech garden-path sentences. To do this, we use both yes–no and open-ended questioning and also test the difference between word-by-word self-paced reading (which does not allow for regressions, but informs us about the presence of the garden-path effect during reading) and whole-sentence presentation (which allows for regressions, but does not inform us about the process of reading).

## Structure of Czech and the current study

Czech is a highly inflectional language. A single suffix (ending) thus expresses several grammatical categories. Czech nouns typically have one ending which specifies their case (nominative, genitive, dative, accusative, vocative, locative, instrumental), gender (masculine, feminine, or neuter), and number (singular or plural). This leads to rather complex paradigms comprising multiple forms. Importantly, some forms may be syncretic (homonymous). For example, the lemma *obchodnice* ('female storekeeper') can have the form *obchodnici*, which can be either dative, accusative, or locative, depending on the context. Morphological syncretism of this type may be easily used for creating locally ambiguous (garden-path) sentences. This can be illustrated with sentences such as (9), which we used in our experiments.

(9) Ostražit-ý        policist-a              prohleda-l         obchodnic-i
    alert-NOM.M.SG policeman-NOM.M.SG search-3SG.M.PST storekeeper-DAT.F.SG
    před          prodejn-ou     dodávk-u     a   odje-l              na
    in front of shop-INST.F.SG van-ACC.F.SG and leave-3SG.M.PST for
    služebn-u.
    station-ACC.F.SG
    'An alert policeman searched a storekeeper's van in front of her shop and left for
    the station.'

The verb *prohledal* ('searched') is transitive, it takes one direct object, and since the following noun, *obchodnici* ('a storekeeper'), corresponds with the accusative case (or dative, or locative), it is analyzed as a patient (since this presents the easiest way of integrating the noun in the sentence at that point of processing). In other words, the initial interpretation is that the policeman searched the storekeeper. However, comprehenders later encounter the noun *dodávku* ('a van'), which is also in the accusative, but non-syncretic (i.e. not homonymous with another form of the noun). As such, it has to be incorporated into the sentence as the direct object of the verb. But the verb *prohledal* already has the position of the direct object filled by the noun *obchodnici* and it cannot take another object. So, the initial analysis has no sufficient attachment site for the newly encountered noun and the sentence has to be reanalyzed, i.e., the noun *obchodnici* has to be reanalyzed as a dative form (it is not possible to analyze it as a locative form, since in Czech, the locative case is strictly prepositional, i.e., a locative form is never used without a preposition requiring a locative complement).

According to the Attach Anyway principle [7], the parser attaches the noun *dodávku* to the position of the object of the verb *prohledal*, even though the result is ungrammatical, and tries to fix the mistake by adjusting the structure. To do this, it needs to detach the noun *obchodnici* from its current position and find its dative interpretation. In Czech, nouns in the dative form can have the function of an indirect object (e.g. recipient) or of an adjunct. In the case of our stimuli, the dative noun is a benefactive or an external possessor (see Fried [27] for more information on external possessivity in Czech). Thus, the parser can reattach the noun to the structure with a different function. The final, faithful and globally coherent interpretation should be that the policeman searched a van that belonged to the storekeeper. The sentence might be difficult to reanalyze because the disambiguating noun does not offer any information (positive

symptoms, in terminology of Fodor [28]) about what needs to be done to fix the structure—there is no cue or symptom that would lead the parser to the dative interpretation of *obchodnici*, the only information is that *dodávku* needs an attachment site and that *prohledal* cannot have two direct objects.

Thus, comprehenders should perform three key steps to interpret the sentence faithfully to the input. They should:

1. Correctly attach the noun *dodávku* to the existing structure as the object of the verb *prohledal*.

2. Detach the ambiguous noun, *obchodnici*, from the object position and deactivate the initial interpretation.

3. Reanalyze the ambiguous noun and reattach it to the structure as a benefactive/possessor.

If the hypothesis of Slattery et al. [21] is correct and a coherent and faithful syntactic structure is built during the reanalysis, comprehenders should be able to perform steps 1 and 3, i.e., to correctly identify the patient and to discover the relationship between the patient and the benefactive/possessor. They should not, however, perform step 2, where they deactivate the initial misinterpretation. Thus, they should arrive at the conclusion that the policeman searched the van and the van belonged to the storekeeper, but the interpretation where the policeman searched the storekeeper should still be active.

We can use simple comprehension questions to test whether comprehenders managed to go through the specific steps during the reanalysis. Namely:

a) *Did the policeman search the van?* for step 1.

b) *Did the policeman search the storekeeper?* for step 2.

c) *Did the storekeeper have a van?* for step 3.

If a comprehender managed to successfully attach the object noun to the structure, their answer to the question a) should be "yes". If they detached the ambiguous noun from its initial position as an object and deactivated the initial misanalysis, the answer to the question b) should be "no". And if they managed to successfully reanalyze the ambiguous noun and to discover its dative interpretation, they should answer "yes" to the last question.

The yes–no version of comprehension questions can influence the way comprehenders respond in several ways. All of the questions inherently suggest a specific interpretation, questions a) and b) can reactivate specific analyses of the sentence through similarities in syntactic structure, and there is also the problem of acquiescence bias [29]—participants have a higher tendency to answer "yes" than "no" to comprehension questions. Since all of this can lead to a higher response accuracy for questions a) and c), and a lower response accuracy for question b) for reasons that are unrelated to sentence processing, we decided to also use open-ended (free recall) questions. Questions a) and b) can be substituted with the question *What did the policeman do?*, and question c) can be changed into *Whose was the van?*. The use of open-ended questions also allows us to perform a qualitative analysis (along with a quantitative one), which may shed more light on the resulting representations of garden-path sentences.

The type of ambiguous nouns that we chose to create the garden-path sentences also allowed us to create non-ambiguous conditions with a very similar meaning, since they have a masculine counterpart that is not ambiguous in its dative form (the dative form is homonymous with the locative form, but nouns in the locative case are preceded by a preposition). This allowed us to create a control condition that only differs from the experimental condition in the grammatical gender of the potentially ambiguous noun, as in sentence (10).

(10) Ostražit-ý policist-a prohleda-l
alert-NOM.M.SG policeman-NOM.M.SG search-3SG.M.PST
obchodník-ovi před prodejn-ou dodávk-u a
storekeeper-DAT.M.SG in front of shop-INST.F.SG van-ACC.F.SG and
odje-l na služebn-u.
leave-3SG.M.PST for station-ACC.F.SG
'An alert policeman searched a storekeeper's van in front of his shop and left for the station.'

Since the noun *obchodníkovi* can only be interpreted as dative in the given context, participants should not be garden-pathed, and there should not be any problems with comprehension caused by temporary ambiguity. There is, however, one more factor that can complicate processing, namely semantics of the disambiguating noun. There are three nouns in every condition, and processing can get more difficult when these nouns are more similar to each other due to encoding interference [30]. Since the ambiguous noun and the object noun already interfere with each other syntactically in the garden-path condition (both can be interpreted as accusative and thus as the direct object of the verb), we decided to control the semantics of the object noun to differentiate between a low interference condition such as (9), where the noun refers to an inanimate object (and thus is semantically less similar to the ambiguous noun), and a high interference condition, where the object noun refers to a person, as in (11). The high interference condition should be more difficult to process, leading to a lower response accuracy in both the garden-path and the control condition.

(11) Ostražit-ý policist-a prohleda-l
alert-NOM.M.SG policeman-NOM.M.SG search-3SG.M.PST
obchodnic-i před prodejn-ou klientk-u a
storekeeper-DAT.F.SG in front of shop-INST.F.SG client-ACC.F.SG and
odje-l na služebn-u.
leave-3SG.M.PST for station-ACC.F.SG
'An alert policeman searched a storekeeper's client in front of her shop and left for the station.'

As our reviewers pointed out, it is also important to note that interference can play a role in information recall. In order to answer the comprehension questions, participants need to reconstruct the previously read sentence and pick the correct response. Doing this might be more difficult when different parts of the sentence are interfering with each other due to semantic or syntactic similarity. This effect might be especially relevant when the initial misanalysis persists. The memory trace it leaves behind can interfere with the new representation, making it more difficult to respond correctly when these two representations are semantically similar to each other. It has also been shown that animate patients are generally considered less acceptable than inanimate ones [31], which might play a role in the decision making process if participants compare the two representations against each other. Since plausibility plays an important role in garden-path sentence processing [5, 6], we can assume that conditions where both representations are similarly plausible or acceptable will be more difficult to interpret correctly than conditions where the correct representation (i.e., the one containing an inanimate patient) is also more acceptable than the initial one.

Overall, this manipulation allows us to create a more diverse set of experimental sentences in terms of difficulty, and lets us explore whether the pattern of responses changes for more problematic sentences.

In sum, we used items consisting of four conditions as shown in Table 1. The same set of 48 items was used in all five experiments.

**Table 1. The structure of items used in all five experiments with glosses.**

| SentType | ObjAnim | Sentence beginning | Ambiguous region | Location | Disambig. region | Sentence end |
|---|---|---|---|---|---|---|
| gp | anim | Ostražitý policista prohledal | *obchodnic-i* | před prodejnou | *klientk-u* | a odjel na služebnu. |
| | | an alert policeman searched | storekeeper-DAT.F.SG = ACC.F.SG | in front of a shop | client-ACC.F.SG | and left for the station |
| | | 'An alert policeman searched the female-storekeeper's client in front of her shop and left for the station.' | | | | |
| gp | inanim | Ostražitý policista prohledal | *obchodnic-i* | před prodejnou | *dod*ávk-u | a odjel na služebnu. |
| | | an alert policeman searched | storekeeper-DAT.F.SG = ACC.F.SG | in front of a shop | van-ACC.F.SG | and left for the station |
| | | 'An alert policeman searched the female-storekeeper's van in front of her shop and left for the station.' | | | | |
| nongp | anim | Ostražitý policista prohledal | *obchodn* ík-ovi | před prodejnou | *klientk-u* | a odjel na služebnu. |
| | | an alert policeman searched | storekeeper-DAT.M.SG | in front of a shop | client-ACC.F.SG | and left for the station |
| | | 'An alert policeman searched the male-storekeeper's client in front of his shop and left for the station.' | | | | |
| nongp | inanim | Ostražitý policista prohledal | *obchodn* ík-ovi | před prodejnou | *dod*ávk-u | a odjel na služebnu. |
| | | An alert policeman searched | storekeeper-DAT.M.SG | in front of a shop | van-ACC.F.SG | and left for the station |
| | | 'An alert policeman searched the male-storekeeper's van in front of his shop and left for the station.' | | | | |

SentType: *gp* = garden-path sentence; *nongp* = non-garden-path sentence.

ObjAnim: *anim* = animate object; *inanim* = inanimate object.

Since the number of verbs that can be plausibly combined with both animate and inanimate patients is limited, we decided to use every verb twice. We also repeated the animate nouns in the disambiguating region for the same reason. The combinations of the verb and the nouns never repeated.

Altogether, five experiments were conducted. Experiment 1 employed an acceptability judgment task and thus aimed to test whether there is indeed a difference between the ambiguous and unambiguous conditions, as well as a difference between the high-interference (animate object) and low-interference (inanimate object) conditions. The four remaining experiments used the self-paced reading task and differed in two important aspects, i.e. presentation mode and questioning type. Experiments 2 and 4 used whole-sentence-at-once presentation self-paced reading, which provided the participants with more natural reading conditions. On the other hand, Experiments 3 and 5 employed the word-by-word presentation mode in order to provide information about reaction times (RTs) on specific regions of the sentence (especially the disambiguating region, i.e. the object noun, where we expected elevated RTs as a result of the garden-path effect in ambiguous sentences). Experiments 2 and 3 used classic yes–no comprehension questions targeting various aspects of the resulting representation, whereas Experiments 4 and 5 employed two open-ended questions, which allowed for a more qualitative analysis of what the resulting representation looks like.

## Data accessibility

Experimental items, data and R scripts used for their analysis are accessible in the following repository: https://osf.io/98g26/.

## Ethics approval

Research ethics board approval for all five experiments was acquired from the Research Ethics Committee of the Faculty of Arts, Charles University (Ref. No.: UKFF/160211/2022–1-1). Participation in the experiments was voluntary, and all participants provided written informed consent. All data used in the current analyses were fully anonymized.

## Experiment 1

As Experiment 1, we ran a pilot study using the acceptability judgment task. Since the structure we worked with has not, to our knowledge, been used in any previous work, and the experimental items were created specifically for the purposes of our experiments, we needed to ensure that the unambiguous versions of the sentences will be considered unproblematic by the readers. If this was not the case, we could not confidently say whether any potential effects on processing and comprehension detected in the next four experiments could be attributed to the garden-path phenomenon, or whether they resulted from the complex nature of the stimuli. We also aimed to see whether we would find the expected difference in acceptability between garden-path and non-garden-path sentences and also between sentences with animate and inanimate objects.

### Method

**Participants.**  We tested 70 undergraduate students of Charles University (54 female and 16 male; mean age = 23.31 years). All participants were native speakers of Czech and participated for course credit.

**Materials.**  Altogether, participants rated the acceptability of 135 sentences. Four sentences were used as practice items at the beginning of the experiment. In the main part of the task, 48 experimental items were used (with four conditions, see Table 1), together with another 96 fillers (24 of which were experimental items for a different experiment).

**Procedure.**  The experiment was conducted online using the IbexFarm platform [32].

First, the participants were informed about the task, namely that they will have to evaluate each sentence using a 5-point scale ranging from absolutely unacceptable ("zcela nepřijatelná") to absolutely acceptable ("zcela přijatelná"). The *absolutely unacceptable* pole was defined as follows: "the sentence is rather disruptive or strange, it can be even unintelligible (or hardly intelligible), and native speakers of Czech would not use it (or only by mistake)". On the other hand, the *absolutely acceptable* pole was defined as follows: "the sentence looks unproblematic, there is nothing surprising or strange about it, and it might be probable to encounter it in spoken or written discourse produced by Czech native speakers".

Second, general demographic information (such as age, gender, and native language) about the respondents were collected.

Third, participants were successively presented with four practice items so that they could get acquainted with the rating process.

Fourth, the main part of the study commenced. The order of experimental items and fillers was randomized for each participant. Only one condition of each experimental item was presented to each participant (based on the Latin-square design).

**Data analysis.**  For the analysis, we used linear mixed-effects models with the lme4 package [33]. The degrees of freedom and *p*-values were estimated using Satterthwaite's approximations from the lmerTest package [34]. As fixed effects, sentence type and object animacy were used (in interaction). Both effects were coded using sum contrasts. For sentence type, GP sentence was coded as 0.5 and non-GP sentence as −0.5. For object animacy, animate object was coded as 0.5 and inanimate object as −0.5. As random effects we used the intercept for participants and items, the by-participants random slope for sentence type and a by-item random slope for animacy.

### Results

The average scores for each condition together with 95% confidence intervals are shown in Table 2.

**Table 2. Acceptability judgments for the experimental items used in this study together with raw mean reaction times in ms (with 95% confidence intervals) for the four conditions in Experiments 2 and 4.**

| SentType | ObjAnim | AJ [95% CI] | RTs Exp2 | RTs Exp4 |
|---|---|---|---|---|
| gp | anim | 2.72 [2.63, 2.82] | 6906 [6756, 7055] | 8282 [8095, 8469] |
| gp | inanim | 3.53 [3.44, 3.63] | 6273 [6140, 6406] | 7627 [7451, 7802] |
| nongp | anim | 3.6 [3.51, 3.68] | 6333 [6197, 6468] | 7801 [7622, 7981] |
| nongp | inanim | 4.16 [4.09, 4.23] | 6051 [5919, 6184] | 7322 [7157, 7487] |

SentType: *gp* = garden-path sentence; *nongp* = non-garden-path sentence.

ObjAnim: *anim* = animate object; *inanim* = inanimate object.

AJ = mean acceptability rating for the given condition (the higher, the more acceptable)

RTs = mean reaction times in ms

square brackets denote 95% confidence intervals of the mean.

The linear-mixed effects model yielded three significant effects: sentence type ($\beta = -0.733$, SE = 0.063, t = $-11.559$, p < 0.001), object animacy ($\beta = -0.695$, SE = 0.059, t = $-11.678$, p < 0.001) and the interaction between sentence type and animacy ($\beta = -0.245$, SE = 0.073, t = $-3.3357$, p < 0.001).

## Discussion

As expected, garden-path sentences were judged as significantly less acceptable than non-garden path sentences. Object animacy also seems to play an important role in the acceptability of a sentence: sentences with animate objects were judged as less acceptable than sentences with inanimate objects. Furthermore, the interaction effect tells us that the difference between sentences with animate and inanimate objects was more pronounced for garden-path sentences. This is consistent with our assumption that sentences with animate patients will be more difficult to interpret due to interference and/or due to the fact that animate nouns are in general less acceptable in the role of a patient or a theme. The interaction with ambiguity also points towards a potential interference of the initial misanalysis and the new analysis of the GP sentences, which complicates the processing even further (as mentioned in the section above).

Thus, the results of the acceptability judgment task show that both the presence of a temporary ambiguity and high interference make comprehenders view sentences as more disruptive and more difficult to comprehend. We can also see that unambiguous sentences score significantly better than their ambiguous counterparts. Because these sentences only differ from the ambiguous ones in grammatical gender of one noun, we can assume that the general structure of the stimuli is not considered problematic by comprehenders and that any potential reading disruptions or decrease in response accuracy found in the ambiguous conditions can be attributed to ambiguity resolution processes.

## Experiment 2

The aim of this experiment was to test whether comprehenders form a full and coherent global syntactic representation of garden-path sentences and whether the initial misinterpretation lingers even after reanalysis is performed due to comprehenders' failure to deactivate it in their memory.

We used intraclausal garden-path sentences with patient/benefactive ambiguity, such as the one in (9). We used three yes–no comprehension questions to probe whether the participants performed all steps required to carry out a successful reanalysis of the sentence and to reach a

faithful final interpretation, namely a) the correct attachment of the disambiguating noun as the patient, b) detaching the ambiguous noun from the object position and deactivating the interpretation based on this syntactic structure, and c) successful reanalysis of the ambiguous noun and reattaching it as a benefactive/external possessor.

The experiment used a self-paced reading paradigm. Sentences were presented at once as a whole to provide the participants with a more natural reading environment. Since rereading previously encountered material is a crucial part of reanalysis processes for many comprehenders, we considered it important not to limit the participants in their ability to perform regressions. Each sentence was followed by a single comprehension question and three options for answer (yes, no, I don't know). The "I don't know" responses were always counted as incorrect and the option was used mainly to allow the readers not to guess the right answer in case they did not know it.

The experiment was preregistered on the Open Science Framework: https://osf.io/vbcp2.

## Hypotheses

Slattery et al. [21] assume that when the initial misanalysis of garden-path sentences lingers, it is caused by a memory trace its semantic representation leaves behind. However, comprehenders should still be able to build a faithful, coherent syntactic representation of a given garden-path sentence during reanalysis. If this is true, we should expect the participants of the current study to successfully attach the disambiguating noun to the existing syntactic structure, to reanalyze the ambiguous noun and reattach it as a benefactive/possessor in the majority of the cases, but also to keep the initial interpretation of the ambiguous noun as a patient active sometimes. Our predictions were thus as follows:

1. Participants will be garden-pathed—we will detect elevated reading times for garden-path (GP) sentences as compared to non-garden-path (non-GP) sentences.

2. The initial misanalysis will linger in some cases—GP sentences will produce significantly more incorrect responses than non-GP sentences for the question targeting the process of detaching the ambiguous noun from the object position or deactivating the initial misanalysis.

3. A global syntactic representation faithful to the input will be formed in the majority of the cases—there will not be a significant difference in response accuracy between GP and non-GP sentences for questions targeting the correct attachment of the disambiguating noun into the object position and the successful reanalysis of the ambiguous noun and its reattachment as a benefactive.
   The last prediction is motivated mostly by methodological concerns that lead us to manipulate the strength of interference caused by the disambiguating noun.

4. Sentences with higher interference (animate objects) will be more difficult to process, leading to elevated RTs and lower response accuracy for all comprehension questions. The effect should be even stronger for garden-path sentences, especially for the questions targeting the initial misanalysis and the correct attachment of the disambiguating noun, since both the initial and the new analysis already interfere syntactically (two nouns compete for the position of the direct object). Inanimate disambiguating nouns might also have an advantage in this competition over animate nouns, because they are slightly more plausible in the role of a patient, making it easier to arrive at the faithful representation of garden-path sentences.

## Method

**Participants.** One hundred seventy four Charles University undergraduate students (147 female and 27 male; mean age = 22.19 years) participated in Experiment 2. All participants were native speakers of Czech and participated for course credit.

**Materials.** Forty-eight experimental items were used in Experiment 2 (see the example of an item in Table 1).

The experiment had a 2x2x3 within-subject design. We manipulated the ambiguity of the sentences by changing the grammatical gender of the nouns in question (feminine vs. masculine). We also manipulated the difficulty of the sentences by changing the animacy of the disambiguating noun (animate vs. inanimate), with animate nouns being semantically more similar to the potentially ambiguous noun (which always referred to a person), thus leading to a stronger interference effect.

We used three yes–no comprehension questions to probe the final representation (see Table 3). The first one focused on the correct attachment of the disambiguating noun (QCOR), the second one on the initial misinterpretation (QMIS) and the third one on the correct interpretation of the ambiguous noun (QPOS). The correct answer to QCOR and QPOS was always "yes", the correct answer to QMIS was always "no". The number of positive and negative responses was balanced through the experiment.

Ninety-six filler items and three practice items were used. Altogether, each participant read 147 sentences. Twenty-eight of the filler sentences were ungrammatical. In eighteen of them, there was a missing or a redundant constituent to mirror the situation of an unfaithful garden-path analysis (if the reanalysis fails, there are two constituents with the same grammatical function in the representation of the sentence and there is no possessor). The remaining ungrammatical fillers were constructed in a different way (in five of them, there was a missing preposition, and in another five, there was an agreement violation). Each filler was followed by a comprehension question. None of the questions targeted a region influenced by the ungrammaticality. For forty of the filler sentences, the correct answer to the comprehension question was "yes", and for the rest, the correct answer was "no" (to balance the overall number of positive/negative responses). For the ungrammatical sentences, the number was also balanced.

**Procedure.** The experiment was web-based and conducted on the IbexFarm platform [32].

At the beginning, participants were informed about the general experimental procedure and they were asked to fill in their gender, age, native language, field of study, and whether they experienced any reading problems such as dyslexia. In the actual experiment, participants first saw a sentence as a series of underscores. Once they pressed the space bar, the sentence appeared. Their task was to read the sentence at their normal reading pace, and once they were finished, they had to press the space bar again, which caused the sentence to disappear and a yes–no comprehension question to appear. Participants responded by clicking the mouse. Once they responded, the next sentence appeared as a series of underscores.

**Table 3. Yes–no comprehension questions used in Experiments 2 and 3.**

| SentType | ObjAnim | Correct response |
|---|---|---|
| QCOR | Did the policeman search the van/the client? | yes |
| QMIS | Did the policeman search the storekeeper? | no |
| QPOS | Did the storekeeper have a van/a client? | yes |

QuesType: *qcor* = question targeting the correct analysis of disambiguating noun; *qmis* = question targeting deactivation of the initial misanalysis; *qpos* = question targeting the correct analysis of the ambiguous noun.

Each participant saw only one of the conditions of each item and the conditions were distributed based on the Latin-square design. Altogether, each participant was exposed to four cases of each condition. The order of sentences was fully randomized for each participant. The experiment took about 20–25 minutes.

**Data analysis.** First, participants' accuracy for filler items was calculated. Only those participants with accuracy scores of 75% or higher were included in subsequent analyses. Fifteen participants scored below 75% and were excluded (these are already not included in the number of participants reported in Participants subsection).

Second, reaction times were trimmed. We excluded RTs lower than 1000 ms, the remaining RTs were log-transformed and the upper cut-off point was set as 2.5 standard deviations from the mean, i.e. 9.928 log(ms) which corresponded to 20 491.67 ms. This resulted in 2.48% of the RTs being excluded.

Third, we used linear-mixed effects regression with the lme4 package [33] to examine the differences in RTs between the conditions. The degrees of freedom and $p$-values were estimated using Satterthwaite's approximations from the lmerTest package [34]. As the dependent variable, we used the log(RTs). As fixed effects, sentence type and object animacy were used (in interaction). Both effects were coded using sum contrasts. For sentence type, GP sentence was coded as 0.5 and non-GP sentence as −0.5. For object animacy, animate object was coded as 0.5 and inanimate object as −0.5. Following the approach by Bates et al. [35], we used the following random effects structure: the intercept for participants and items, the by-participants random slope for sentence type and a by-item random slope for sentence type. For this analysis, we report the beta estimates, standard errors (SEs), $t$-values, and $p$-values.

Fourth, response accuracy was analyzed using logit mixed-effects models [36] with question type as a fixed effect, sentence type and object animacy as nested effects within question type (in interaction), and intercept for participant and item as random effects (with question type as a random slope both for items and for participants) which again followed recommendations by Bates et al. [35]. This allowed us to directly compare general response accuracy for the three question types and also to examine the animacy and sentence type effects for each question type separately. Question type was coded using Helmert contrasts [37] and the contrast matrix is shown in Table 4. Contrast 1 targeted the difference between questions targeting the real object (QCOR) and the question targeting the possessive relationship (QPOS). Contrast 2 targeted the difference between the mean of the two questions (QCOR and QPOS) and the question targeting the initial misanalysis (QMIS). Sentence type and object animacy were coded using sum contrasts (as in the RTs analysis). For this analysis, the beta estimates, standard errors (SEs), z-scores, and $p$-values are reported (only for significant results).

Fifth, an analysis of the incorrect response types was conducted. This analysis was supplementary to the analysis of response accuracy and due to the generally low number of incorrect responses, it is rather descriptive and qualitative.

**Table 4. Helmert contrasts matrix for the question type variable used in the analysis of response accuracy in Experiments 2 and 3.**

| Question type | Contrast 1 | Contrast 2 |
|---|---|---|
| QCOR | 1 | -1 |
| QMIS | 0 | 2 |
| QPOS | -1 | -1 |

QuesType: *qcor* = question targeting the real object; *qmis* = question targeting the initial misanalysis; *qpos* = question targeting the possessive relationship.

## Results

**Reaction times.**  Raw reaction times for each condition are presented in Table 2. The linear mixed-effects model [33] showed significant effects for sentence type ($\beta = 0.062$, SE = 0.016, t = 5.345, p < 0.001), object animacy ($\beta = 0.069$, SE = 0.008, t = 8.469, p < 0.001) and also for the interaction between the fixed effects ($\beta = 0.05$, SE = 0.016, t = 3.032, p < 0.01).

**Response accuracy.**  The overall response accuracy for participants was relatively high. Participants' accuracy scores ranged from 44.68% to 100%, the average accuracy being 88.32% (median 89.58%). The between-item variability was 73.41% to 96.51% (mean 88.39%, median 88.19%).

Table 5 and Fig 1 show the response accuracy for each condition. The logit mixed-effects model revealed several significant effects. For question type, there was a significant effect of Contrast 2 ($\beta = -0.237$, SE = 0.039, z = −6.019, p < 0.001), meaning that the response accuracy for questions targeting the initial misanalysis was significantly lower than the mean accuracy for questions targeting the true object and questions targeting the possessive relationship. For questions targeting the true object, the model yielded significant effects of sentence type ($\beta = -0.572$, SE = 0.144, z = −3.968, p < 0.001) and object animacy ($\beta = -0.621$, SE = 0.144, z = −4.315, p < 0.001). For questions targeting the initial misanalysis, we found only a significant effect of sentence type ($\beta = -1.342$, SE = 0.119, z = −11.25, p < 0.001). And for questions targeting the possessive relationship, we found significant effects of object animacy ($\beta = -0.766$, SE = 0.144, z = −5.329, p < 0.001) and the interaction between sentence type and object animacy ($\beta = -0.795$, SE = 0.287, z = −2.772, p < 0.01).

## Discussion

We replicated the finding of Christianson et al. [11]. The initial misanalysis of the ambiguous region did often linger even in the case of our sentences—the participants gave incorrect responses to the question targeting the initial misanalysis (*Did the policeman search the*

**Table 5. Response accuracy in Experiments 2 and 3 which employed yes–no questions.**

| SentType | ObjAnim | QuesType | Exp-whole | | Exp-wbw | |
|---|---|---|---|---|---|---|
| | | | N | % inc | N | % inc |
| gp | anim | qcor | 95/688 | 13.8% | 95/633 | 15.01% |
| gp | anim | qmis | 190/691 | 27.5% | 223/633 | 35.23% |
| gp | anim | qpos | 90/694 | 13% | 90/633 | 14.22% |
| gp | inanim | qcor | 55/689 | 7.98% | 57/634 | 8.99% |
| gp | inanim | qmis | 152/692 | 22% | 167/633 | 26.38% |
| gp | inanim | qpos | 34/658 | 4.91% | 52/633 | 8.21% |
| nongp | anim | qcor | 58/694 | 8.36% | 65/634 | 10.25% |
| nongp | anim | qmis | 66/693 | 9.52% | 63/632 | 9.97% |
| nongp | anim | qpos | 72/692 | 10.4% | 65/632 | 10.28% |
| nongp | inanim | qcor | 34/689 | 4.93% | 29/635 | 4.57% |
| nongp | inanim | qmis | 65/694 | 9.37% | 76/635 | 11.97% |
| nongp | inanim | qpos | 53/691 | 7.67% | 54/633 | 8.53% |

SentType: *gp* = garden-path sentence; *nongp* = non-garden-path sentence.

ObjAnim: *anim* = animate object; *inanim* = inanimate object.

QuesType: *qcor* = question targeting the real object; *qmis* = question targeting the initial misanalysis; *qpos* = question targeting the possessive relationship.

N = number of incorrect answers / total number of answers.

% inc = percentage of incorrect answers.

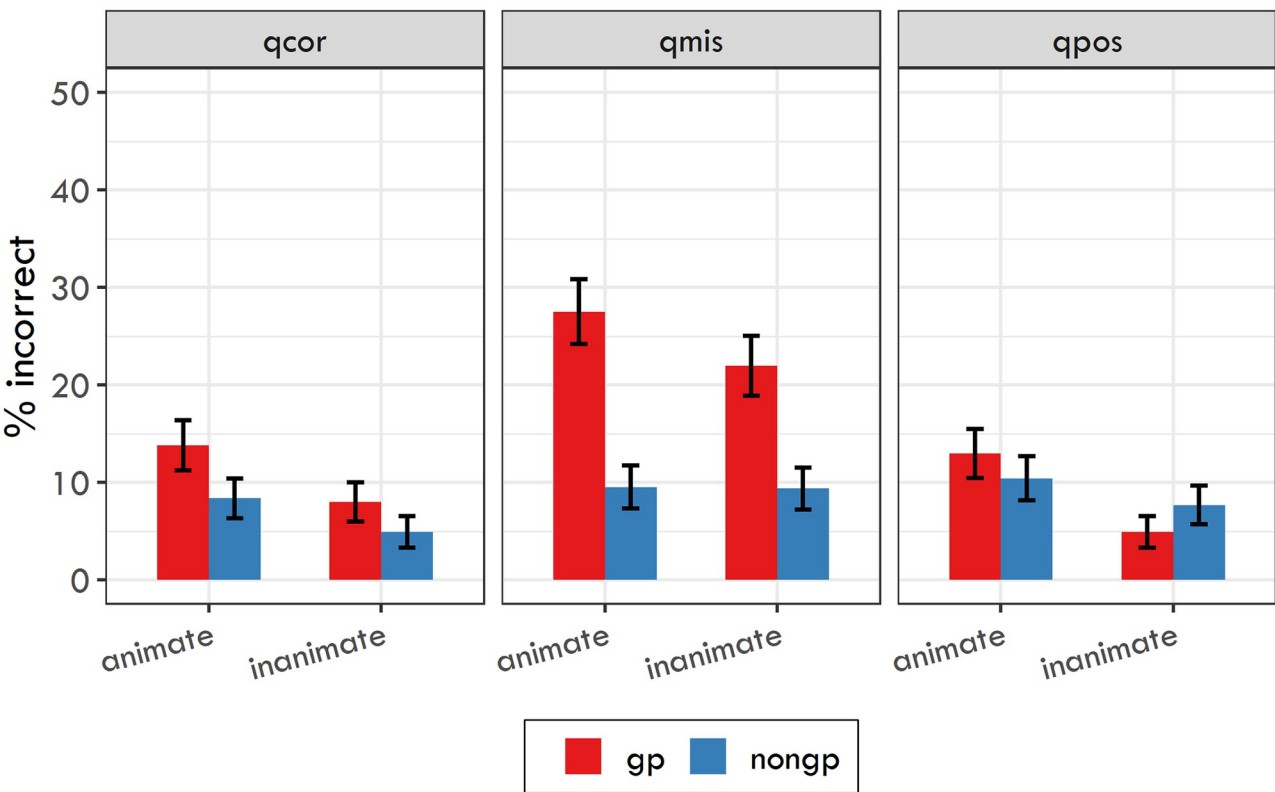

**Fig 1. Response accuracy for the three question types used in Experiment 2 with 95% confidence intervals.** QCOR = question targeting the real object; QMIS = question targeting the initial misanalysis; QPOS = question targeting the possessive relationship.

*storekeeper?*) significantly more often when the question followed garden-path sentences than when it followed unambiguous sentences. This question also caused more difficulties in comparison with the other two comprehension questions targeting other regions of the sentence.

Our results also seem to be largely consistent with the account of Slattery et al. [21]. The response accuracy for both questions targeting the real object (QCOR) and questions targeting the possessive relationship (QPOS) was high, especially compared to questions targeting the initial misanalysis (QMIS). We can thus assume that in the majority of cases, comprehenders managed to correctly attach the object to the structure and successfully reanalyze the ambiguous noun. We also did not find any significant difference between GP and non-GP sentences for QPOS, which supports the predictions we made based on the conclusions of Slattery et al. [21]. We did, however, find a significant difference for QCOR, suggesting that on some occasions, comprehenders had difficulties interpreting the object noun faithfully in GP sentences.

It is possible that, in these instances, the reanalysis fails. Comprehenders do not manage to build a coherent representation of the whole sentence and they stick to the (locally licit) initial interpretation instead. This scenario should lead to a similar difference in QPOS, since the dative interpretation of the ambiguous noun should not have been discovered. However, it is important to note that the way QPOS is phrased, i.e., *Did the storekeeper have a van?*, makes the question very trivial, since it allows the participants to rely on general inference to answer the question correctly. It introduces a plausible relation between the ambiguous noun (which is still interpreted as the grammatical object) and the disambiguating noun, which might not have been attached to the structure at all. If the participants did not form any coherent

representation for the second part of the first clause or were unsure about it, they could still answer the question correctly based on inference only. If this is the case, participants should be significantly less successful in responding to QPOS following after garden-path sentences when the question is open-ended and does not directly suggest a relationship between a possessor and a possessum (e.g., *Whose was the van?*).

We also documented significant effects of difficulty (object animacy) for all questions except for QMIS, regardless of sentence type. We will return to this in the General Discussion.

Longer RTs for garden-path sentences suggest that participants faced processing difficulties while reading the sentences. The mode of presentation, however, does not allow us to be sure that these difficulties were caused by the garden-path effect, since we cannot locate the specific regions of the sentence that were problematic for the participants. To do this, we decided to run another experiment in the word-by-word presentation mode.

The elevated RTs for sentences with higher difficulty also show that the similarity between elements of the sentence causes processing difficulties, and the interaction with sentence type suggests that interference might also further complicate garden-path sentence processing.

## Experiment 3

Experiment 3 was a replication of Experiment 2 with one important difference. The stimuli were not presented in a sentence-at-once mode, but using a classic word-by-word presentation. Such presentation provide us with important information about the course of processing, which will be crucial for the interpretation of our results. Measuring RTs on specific regions of the sentence can also give us information about additional difficulty in processing that relates to the changes of animacy of the disambiguating noun.

The experiment was preregistered on the Open Science Framework: https://osf.io/c5wby.

### Hypotheses

The hypotheses regarding comprehension accuracy were the same as for Experiment 2.

We also expected to find elevated RTs on the disambiguating region (the object noun) and potential spillover regions for garden-path sentences as evidence of an ongoing reanalysis. Encoding interference effects should also lead to elevated reading times on the same regions in high difficulty sentences. Sentences with animate patients should also lead to elevated RTs on the disambiguating region, either due to encoding interference or the lower plausibility of animate nouns as patients. If the animacy further complicates garden-path sentence reanalysis because participants find it more difficult to pick the correct analysis when the two potential patients are semantically more similar, we can also expect the animacy to interact with ambiguity.

### Method

**Participants.**   One hundred fifty nine Charles University undergraduate students (129 female, 28 male, and two did not want to state; mean age = 23.65 years) participated in Experiment 3. All participants were native speakers of Czech and participated for course credit.

**Materials.**   We used the same materials as in Experiment 2.

**Procedure.**   The procedure was very similar to Exp 1. The only difference was that this time, word-by-word self-paced reading was used instead of the whole sentence presentation mode. Each sentence was thus presented as a series of underscores and each spacebar press revealed just one word (and simultaneously hid the previous one).

**Data analysis.** The same analytic steps were followed as in Experiment 2. The only difference stemmed from the fact that this time, the word-by-word presentation mode was employed.

Two participants were excluded based on their less than 75% response accuracy for filler items (these are not included in the sample size reported above).

Data trimming followed these steps. Reaction times below 150 ms were excluded from the analysis. The remaining reaction times were log-transformed and the upper boundary was set as 2.5 standard deviations from the mean, i.e. 7.3 log(ms) or 1486 ms. Altogether, 2.88% of the reaction time data was excluded. The same contrast coding was used for the linear mixed-effects modeling as in Experiment 2. Reaction times were analyzed for separate regions. We were interested in the disambiguating region (i.e. the real object) and the two following regions (to assess the potential spillover effects). Secondarily, we also analyzed the ambiguous region and two following regions (to assess whether the processing of the sentence prior to the disambiguating region was different for the two sentence types). The analysis for the disambiguating region used sentence type and object animacy as fixed effects in interaction, the analysis for the ambiguous region used only the sentence type as a fixed effect (because the object animacy has not yet manifested itself in this part of the sentence). In both analyses, random slopes for each region were determined following the approach by Bates et al. [35].

The analysis of response accuracy and the qualitative analysis of the incorrect answers were identical as in Experiment 2, i.e., question type was used as the main fixed effect, and sentence type and object animacy were used as nested effects within question type (in interaction).

## Results

**Reaction times.** Fig 2 shows RTs in ms for each sentence region used in Experiment 3. The analysis targeted the disambiguating region 7 (i.e., the real object) and the two following regions (i.e., 8 and 9). For all three regions sentence type was used as a random slope for participants. For regions 7 and 9, object animacy was included as a random slope for items and for region 8, it was used as a random slope for participants together with sentence type (without interaction). The pattern of results was practically identical for these regions. For region 7, the model yielded separate significant effects for sentence type ($\beta = 0.035$, SE = 0.008, t = 4.256, $p < 0.001$) and object animacy ($\beta = 0.028$, SE = 0.012, t = 2.342, $p < 0.05$), but no interaction effect. The same results were documented for region 8: sentence type ($\beta = 0.075$, SE = 0.008, t = 9.817, $p < 0.001$), object animacy ($\beta = 0.039$, SE = 0.006, t = 6.048, $p < 0.001$), and also for region 9: sentence type ($\beta = 0.047$, SE = 0.007, t = 7.19, $p < 0.001$), object animacy ($\beta = 0.022$, SE = 0.007, t = 2.989, $p < 0.01$).

The secondary analysis targeted the ambiguous region 4 and the two following regions (i.e. 5, and 6). For these regions, no significant effect of sentence type was found, meaning that the unambiguous dative and ambiguous dative/accusative nouns did not differ in their processing.

**Response accuracy.** The overall response accuracy for participants was again relatively high. Participants' accuracy scores ranged from 52.08% to 100%, the average accuracy being 86.36% (median 89.58%). The between-item variability was 77.22% to 92.36% (mean 86.37%, median 87.34%).

Table 5 and Fig 3 show the response accuracy for each condition. The logit mixed-effects model again revealed several significant effects. As in Experiment 2, there was a significant effect of Contrast 2 for question type ($\beta = -0.293$, SE = 0.041, z = -7.126, $p < 0.001$). In other words, response accuracy for questions targeting the initial misanalysis was significantly lower than the mean accuracy for questions targeting the true object and questions targeting the possessive relationship. For questions targeting the true object, the model yielded significant

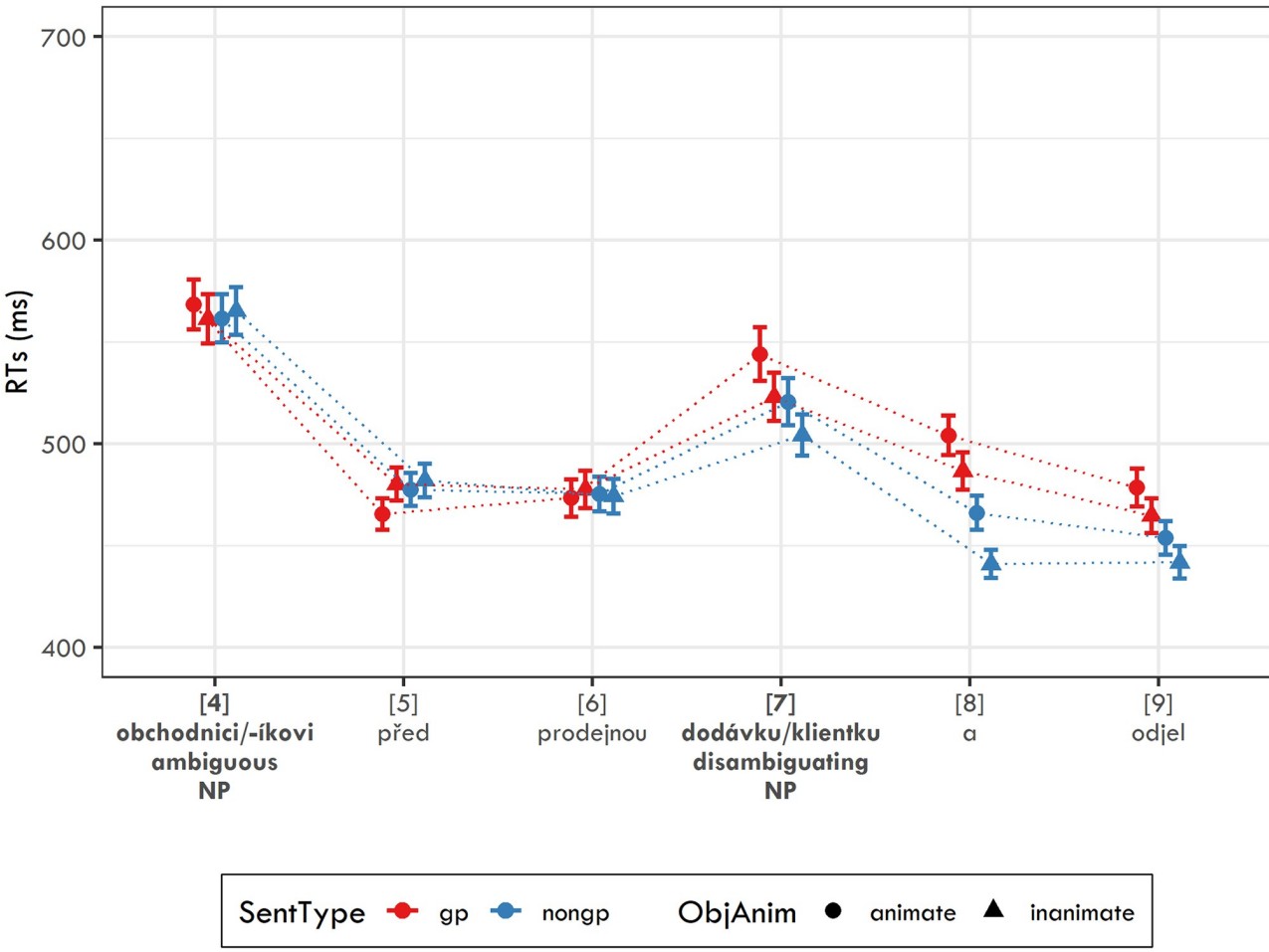

**Fig 2. Reaction times in ms for experimental sentences used in Experiment 3 together with their 95% confidence intervals.**

effects of sentence type ($\beta$ = −0.622, SE = 0.147, z = −4.23, p < 0.001) and object animacy ($\beta$ = −0.783, SE = 0.148, z = −5.308, p < 0.001). For questions targeting the initial misanalysis, we found significant effects of sentence type ($\beta$ = −1.489, SE = 0.118, z = −12.603, p < 0.001) and of the interaction between sentence type and object animacy ($\beta$ = −0.758, SE = 0.23, z = −3.291, p < 0.001). And for questions targeting the possessive relationship, we found a significant effect of object animacy ($\beta$ = −0.447, SE = 0.137, z = −3.264, p < 0.01).

## Discussion

The analysis of RTs revealed significant effects of sentence type on the disambiguating region and the two following regions. Longer RTs on the disambiguating region in the case of garden-path sentences provide evidence of participants being garden-pathed and facing processing difficulties when encountering an element that does not fit into their initial representation of the sentence. The slow-down on the two following regions can be attributed to spillover effects.

In line with our predictions, we also found a similar pattern for high difficulty sentences. The semantic similarity between the ambiguous and the disambiguating noun caused the processing to slow down when the second noun was encountered. Once again, we detected

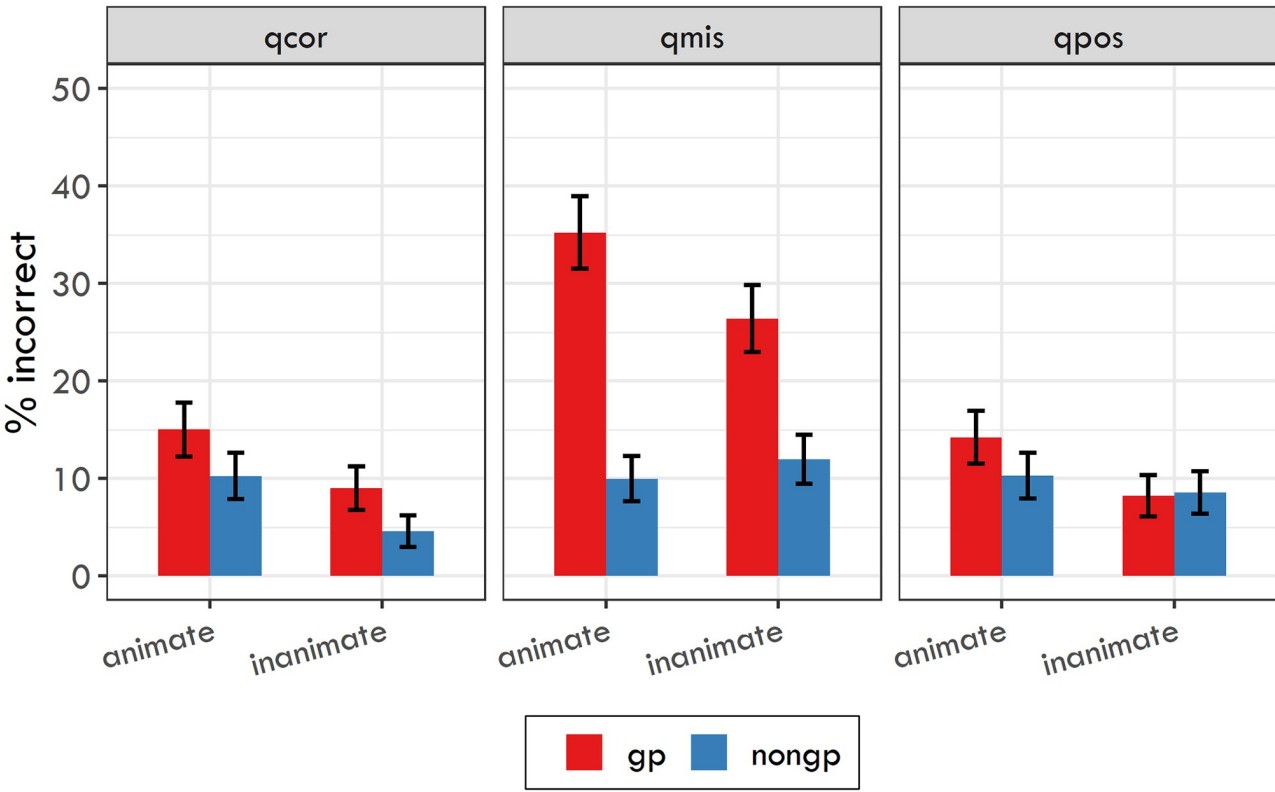

**Fig 3. Response accuracy for the three question types used in Experiment 3 with 95% confidence intervals.** QCOR = question targeting the real object; QMIS = question targeting the initial misanalysis; QPOS = question targeting the possessive relationship.

spillover effects on the two following regions. However, the effect of interference on these regions was independent from the effect of ambiguity, suggesting that the animacy does not directly influence the reanalysis process.

The analysis of response accuracy replicated the findings of Experiment 2. Once again, we found significant evidence for lingering misanalysis. The overall response accuracy for questions targeting the initial misanalysis (QMIS) was significantly lower than for the other two comprehension questions, and the participants produced significantly more incorrect responses when the QMIS followed after garden-path sentences. The difference between garden-path and non-garden-path sentences was even more pronounced in Experiment 3, showing that the word-by-word presentation mode strengthened the lingering misanalysis effect.

For questions targeting the correct attachment of the disambiguating noun to the structure and the correct reanalysis of the ambiguous noun, the pattern of results was identical to Experiment 2. The overall response accuracy was high for both questions targeting the real object (QCOR) and questions targeting the possessive relationship (QPOS), suggesting that, in most of the cases, comprehenders were able to analyze the sentences faithfully to the input. However, we once again found an effect of the sentence type for QCOR, meaning that the participants had more difficulties answering the comprehension question correctly when it followed after garden-path sentences. No such effect was found for QPOS (a possible explanation of this situation was mentioned in the discussion of Experiment 2).

As in Experiment 2, we also found significant effects of sentence difficulty (object animacy) for both QCOR and QPOS. Combined with the elevated RTs, we can assume that semantic

similarity of the two nouns complicates sentence processing and makes it more difficult to recall information from these sentences. In addition, we found an interaction between interference and sentence type for QMIS, meaning that high interference makes it more difficult to interpret the garden-path sentences coherently and in accordance with the input. But since we found no such interaction on RTs, it might be a post-processing effect, e.g., participants find it more difficult to reconstruct the sentence correctly when answering the comprehension questions.

The results so far suggest that comprehenders are more successful in analyzing Czech garden path sentences of this type (as compared to the structure used by Chromý [20]) and that reanalysis often results in a coherent and faithful global representation of the sentence. But there are also a substantial number of cases where reanalysis fails and comprehenders maintain the initial partial analysis of the sentence rather than build a global representation.

These results can however be influenced by the chosen form of comprehension questions. As mentioned before, yes–no questions as a tool of probing the representation of given sentences have various disadvantages. First, participants have a high chance of guessing the answer even if their resulting representation of the sentence offers only insecure cues for the answer. Second, yes–no comprehension questions have—by their nature—to contain certain cues for answering and reactivating sentence content. For example, misanalysis of well-known garden-path sentences such as *While Anna dressed the baby played in the crib* has been targeted by questions such as *Did Anna dress the baby?*, which repeats significant portions of the previously read sentence and may be considered as leading (cf. an open-ended question such as *What did Anna do?*). Third, yes–no questions do not enable a deeper, qualitative analysis of the results and therefore offer us only coarse information about the nature of resulting (mis) representations.

To address the main goals of the paper, we thus decided to run two additional experiments using open-ended questions.

## Experiment 4

The aim of Experiment 4 was to use open-ended questions to probe the final representation of given garden-path sentences. Experiment 4 is a replication of Experiment 2. It employs the same experimental design (self-paced reading with sentences presented at once) and the same set of stimuli, but instead of three yes–no comprehension questions, we used two open-ended questions targeting the regions of interest in the sentence.

The first question, QWHAT, tested which of the two nouns the comprehenders chose to attach to the verb as its object, e.g., *What did the policeman do?* Because the verb in the target sentence is transitive, the response needs to contain an object in the accusative to be grammatical and informative. The response can provide us with information about whether participants stayed committed to the initial interpretation, or whether they correctly attached the disambiguating noun into this position (or whether they maintained both analyses at the same time). The second question, QWHOSE, targeted the faithful (dative) interpretation of the ambiguous noun via its relation to the disambiguating noun, e.g., *Whose was the van?*. The responses were manually coded for the correctness (yes/no) and type of error.

The experiment was preregistered on the Open Science Framework: https://osf.io/km3ga.

### Hypotheses

The hypotheses are similar to previous experiments. We expect to detect longer RTs for garden path sentences and for sentences with animate patients due to (mutually unrelated) processing

difficulties caused by the participants being garden-pathed and by higher interference or lower acceptability of animate nouns as patients.

We also expect the initial misanalysis to linger even after the reanalysis was performed. The question that targets the object of the sentence, e.g., *What did the policeman do?*, should produce more incorrect responses when it follows after garden-path sentences than when it follows after their unambiguous counterparts. These responses should also show that the initial misinterpretation is still active, e.g., *he searched the storekeeper*.

If a full, coherent and faithful global syntactic structure is built during the reanalysis, there should not be a significant difference in response accuracy between the ambiguous and unambiguous sentences for the question targeting the dative interpretation of the ambiguous noun, e.g., *Whose was the van?*.

We also expect to find an effect of difficulty for both comprehension questions. Sentences with animate objects should produce more incorrect responses to both questions regardless of ambiguity. Since we found an interaction of animacy and ambiguity in the last experiment as a potential result of post-processing interference, we can expect to find a similar effect here, with the question targeting the object of the sentence producing the most incorrect responses when the sentence contains a local ambiguity and an animate disambiguating noun.

## Method

**Participants.**   One hundred seventy nine Charles University undergraduate students (152 female, 24 male, and 3 did not want to state; mean age = 22.89 years) participated in Experiment 4. All participants were native speakers of Czech and participated for course credit.

**Materials.**   The same set of sentences was used as in Experiments 2 and 3. This time, however, open-ended questions were used for all stimuli. Only two question types were used this time: QWHAT, which targeted the object of the sentence (and thus tested whether the misanalysis lingered), and QWHOSE, which targeted the faithful, dative interpretation of the ambiguous noun.

**Procedure.**   The procedure was almost identical to Experiment 2. The only difference was that each sentence was followed by an open-ended question instead of a yes–no question. The participants' task was to answer such questions by typing and pressing enter.

**Data coding.**   Since the experiment used open-ended questions and the participants were not limited in any way while answering them, we collected a wide variety of responses. The responses were manually coded by the researchers. To provide consistency, we created a set of instructions for the coding (we asked three independent annotators to code the responses in a pilot experiment and used the annotated results to finish the manual). We used two levels of annotation. The first level signaled whether the response is correct or incorrect. For a response to be considered correct, it had to be semantically and syntactically correct, it could repeat or paraphrase relevant parts of the original sentence and it had to be informative and involve enough detail. For the question targeting the object, QWHAT (*What did the policeman do?*), we only considered as correct those responses that directly and explicitly mentioned the patient (e.g., *he searched the van*). The second level of annotation was used to categorize incorrect responses according to the type of mistake the participants made. We used the following four categories of incorrect responses:

a) "I don't know"—the participant explicitly stated that they do not know the answer to the question or that the information we asked about was not stated in the sentence.

b) "substitution"—responses which contained a sensible, but incorrect answer to the given question (e.g., *What did the policeman do?—He searched the station.*). We also documented what portion of the responses belonging to this category corresponded directly with the

initial misanalysis, i.e., what portion of the responses stated the ambiguous noun as the patient (*He searched the storekeeper.*). These cases were coded as "GP".

c) "different"—the participant seemingly answered a different question (e.g., *What did the policeman do? –In front of the shop.*)

d) "other"—this category included incomplete and underspecified responses (*He searched someone.*) or responses that did not fit into any of the previous categories (responses that were impossible to interpret due to typing errors etc.)

The model example of different types of incorrect responses to the question *What did the policeman do?* can be found in S1 Table).

**Data analysis.** The same analytical steps were taken as in Experiment 2. All participants scored above 75% in response accuracy for filler items, meaning that no one was excluded based on this criterion. RTs were trimmed using 1000 ms as a lower cut-off point and the upper cut-off point was set based as 2.5 standard deviations from the mean log(RT), i.e. 10.149 log(ms) which corresponded to 25 576.49 ms. The linear mixed-effects model had the same structure as in Experiment 2; the only difference was that this time, sentence type and object animacy were used as random slopes for items and no random slope was used for participants due to singularity issues.

Response accuracy and incorrect answers were analyzed similarly as in the previous experiments, i.e., using a nested analysis with question type as the main fixed effect, and with sentence type and animacy as the fixed effects nested within question type. The only difference stemmed from the fact that question type had only two values: QWHAT and QWHOSE. Therefore, we employed sum contrast coding with QWHAT coded as 0.5 and QWHOSE as −0.5.

## Results

**Reaction times.** Raw reaction times for each condition are presented in Table 2. The linear mixed-effects model [33] showed significant effects for sentence type ($\beta = 0.049$, SE = 0.012, t = 4.032, p < 0.001), object animacy ($\beta = 0.069$, SE = 0.014, t = 4.897, p < 0.001). In other words, an identical pattern of results has been found as in Experiment 2, except the interaction between the fixed effects has not reached significance this time.

**Response accuracy.** The overall response accuracy for participants was lower than in both of the experiments using yes–no questions. Participants' accuracy scores ranged from 22.92% to 100%, the average accuracy being 80.09% (median 83.33%). The between-item variability was 65.92% to 90.5% (mean 80.09%, median 79.61%).

Table 6 and Fig 4 show the response accuracy for each condition. The logit mixed-effects model revealed several significant effects. There was a significant effect of question type ($\beta = 0.206$, SE = 0.077, z = -2.695, p < 0.01). For questions targeting the initial misanalysis, the model yielded significant effects of sentence type ($\beta = −0.702$, SE = 0.087, z = −8.07, p < 0.001) and object animacy ($\beta = −0.654$, SE = 0.087, z = −7.518, p < 0.001). And for questions targeting the possessive relationship, we found significant effects of object animacy ($\beta = −0.661$, SE = 0.082, z = −8.081, p < 0.001) and the interaction between sentence type and object animacy ($\beta = −0.393$, SE = 0.163, z = −2.419, p < 0.05).

**Incorrect responses.** The frequency of the four different types of incorrect answers is shown in Table 7. It can be seen that the Different type (the case in which the participants seemingly answered a different question) occurred only rarely. The three other types were more frequent. For the question targeting the initial misanalysis, QWHAT, the biggest number of incorrect responses belonged to the Substitution category and to the "Other" category. For the question targeting the possessive interpretation of the ambiguous noun, the most frequently

**Table 6. Response accuracy in Experiments 4 and 5, which employed open-ended questions.**

| SentType | ObjAnim | QuesType | Exp-whole | | Exp-wbw | |
| --- | --- | --- | --- | --- | --- | --- |
| | | | N | % inc | N | % inc |
| gp | anim | qwhat | 315/1074 | 29.33% | 263/842 | 31.24% |
| gp | anim | qwhose | 299/1074 | 27.84% | 286/844 | 33.89% |
| gp | inanim | qwhat | 182/1074 | 16.95% | 181/844 | 21.45% |
| gp | inanim | qwhose | 169/1074 | 15.74% | 174/842 | 20.67% |
| nongp | anim | qwhat | 176/1074 | 16.39% | 171/842 | 20.31% |
| nongp | anim | qwhose | 261/1074 | 24.3% | 260/842 | 30.88% |
| nongp | inanim | qwhat | 119/1074 | 11.08% | 88/840 | 10.48% |
| nongp | inanim | qwhose | 190/1074 | 17.69% | 170/844 | 20.14% |

SentType: *gp* = garden-path sentence; *nongp* = non-garden-path sentence.

ObjAnim: *anim* = animate object; *inanim* = inanimate object.

QuesType: *qwhat* = question targeting the initial misanalysis; *qwhose* = question targeting the possessive relationship.

N = number of incorrect answers / total number of answers.

% inc = percentage of incorrect answers.

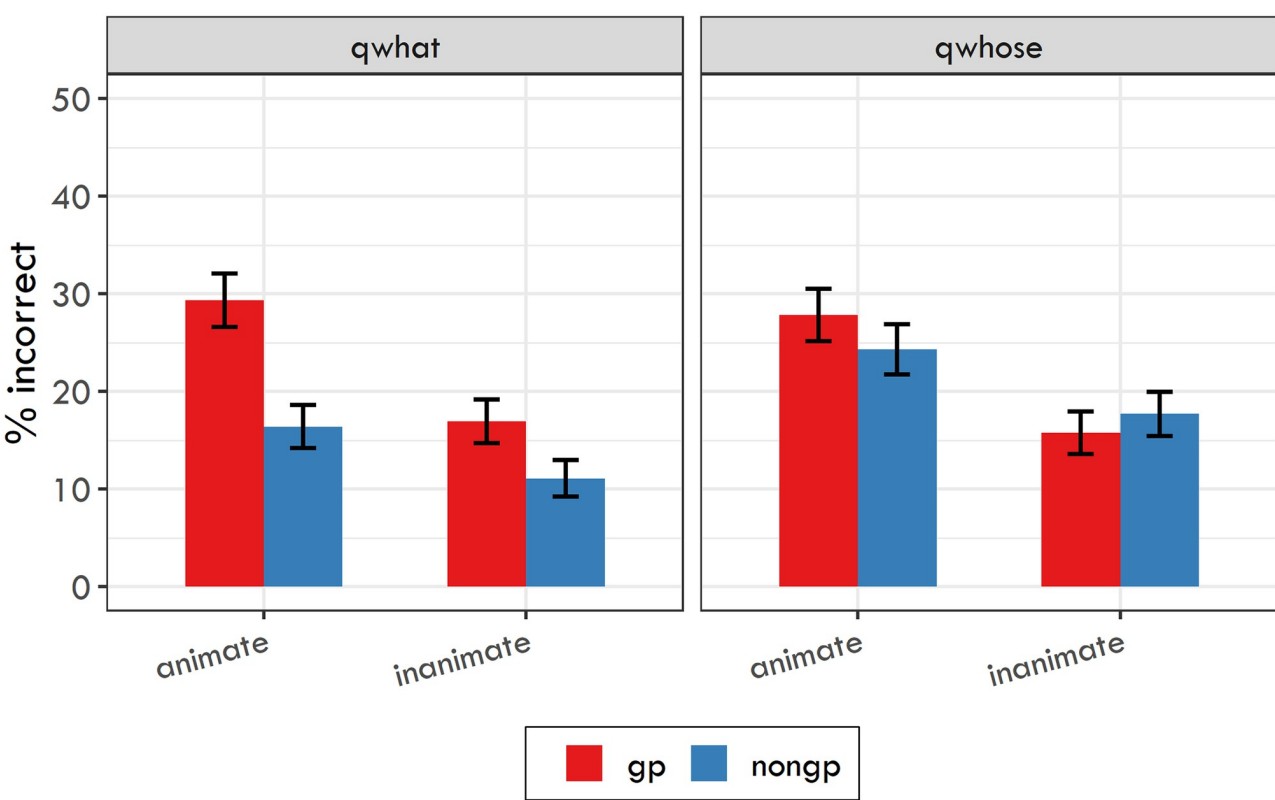

**Fig 4. Response accuracy for the two question types used in Experiment 4 with 95% confidence intervals.** *qwhat* = question targeting the initial misanalysis; *qwhose* = question targeting the possessive relationship.

**Table 7. Frequency of the four types of incorrect responses in Experiment 4.**

| SentType | ObjAnim | QuesType | Error type | | | |
|---|---|---|---|---|---|---|
| | | | I don't know | Different | Substitution | Other |
| gp | anim | qwhat | 29 (9.21%) | 0 | 194 (61.59%) | 92 (29.21%) |
| gp | anim | qwhose | 195 (65.22%) | 4 (1.34%) | 89 (29.77%) | 11 (3.68%) |
| gp | inanim | qwhat | 17 (9.34%) | 0 | 93 (51.1%) | 72 (39.56%) |
| gp | inanim | qwhose | 94 (55.62%) | 3 (1.78%) | 64 (37.87%) | 8 (4.73%) |
| nongp | anim | qwhat | 22 (12.5%) | 0 | 72 (40.91%) | 82 (46.59%) |
| nongp | anim | qwhose | 155 (59.39%) | 2 (0.77%) | 95 (36.4%) | 9 (3.45%) |
| nongp | inanim | qwhat | 18 (15.13%) | 0 | 40 (33.61%) | 61 (51.26%) |
| nongp | inanim | qwhose | 95 (50%) | 1 (0.53%) | 90 (47.37%) | 4 (2.11%) |

SentType: *gp* = garden-path sentence; *nongp* = non-garden-path sentence.

ObjAnim: *anim* = animate object; *inanim* = inanimate object.

QuesType: *qwhat* = question targeting the initial misanalysis; *qwhose* = question targeting the possessive relationship.

represented category was "I don't know", followed by "Substitution". This pattern was similar for all four conditions.

We also performed a more detailed analysis of the types of responses within the Substitution category. The number of responses that were consistent with the initial misanalysis (i.e., when the question *What did the policeman do?* elicited responses like *he searched the storekeeper*, *he searched the storekeeper or the van*, and *he searched the storekeeper and the van*) can be seen in Table 8.

As we can see, garden-path sentences clearly elicited more responses that were consistent with the potential initial misanalysis than non-garden-path sentences. The given type of error also formed more than 50% of the Substitution category for both the high- and low-interference garden-path sentences.

## Discussion

The analysis of RTs revealed the same pattern as in Experiment 2. The participants needed more time to read garden-path sentences and sentences with animate objects. These results can be attributed to processing difficulties caused by the participants being garden-pathed and by interference effects or lower plausibility of animate patients. Similar to Experiment 2, we also observed an interaction of ambiguity and animacy, suggesting that the presence of animate patients can further complicate the processing of garden-path sentences, perhaps by

**Table 8. Frequency of the GP error type in response to *qwhat* in Experiments 4 and 5.**

| SentType | ObjAnim | Experiment 4 | | Experiment 5 | |
|---|---|---|---|---|---|
| | | Substitutions | GP errors | Substitutions | GP errors |
| gp | anim | 194 | 138 (71.13%) | 141 | 89 (63.12%) |
| gp | inanim | 93 | 58 (62.37%) | 88 | 59 (67.05%) |
| nongp | anim | 72 | 5 (6.94%) | 58 | 3 (5.17%) |
| nongp | inanim | 40 | 4 (10.00%) | 30 | 2 (6.66%) |

SentType: *gp* = garden-path sentence; *nongp* = non-garden-path sentence.

ObjAnim: *anim* = animate object; *inanim* = inanimate object.

making it more difficult for comprehenders to identify the correct interpretation of the sentence.

The overall response accuracy was lower than in previous experiments, which is to be expected given the change in the form of the comprehension question. It has been shown before that the form of the comprehension question can influence the form of the response [25] and even the course of processing [38]. For example, according to Ozuru et al. [39], using open-ended comprehension questions can lead to a lower response accuracy as compared to multiple-choice questions.

Since the form of both of the questions was qualitatively different, we cannot directly compare the results to Experiment 2 and 3. We can, however, compare the general information we get about the potential representation of the sentence.

Once again, we found evidence for lingering misanalysis. The question asking about the object of the sentence (i.e. the correct attachment of the disambiguating noun and reanalysis of the ambiguous noun) produced more incorrect responses when it followed garden-path sentences. More importantly, this difference can be attributed to responses that are consistent with the initial misanalysis (e.g., *the policeman searched the storekeeper*). Responses of other types (I don't know, other and even other responses of the substitution type) were evenly distributed across both the garden-path and non-garden-path conditions. This pattern strongly suggests that the initial misanalysis has not been deactivated after the whole sentence was read or that it was somehow reactivated again when participants responded to the question. These results are once again in line with previously mentioned accounts of garden-path sentence processing, e.g., [11–13, 26]. However, we would like to note that we also detected a significant number of cases where the respondents were clearly unsure about the meaning of the sentence or formed a grammatically incorrect representation, e.g., *the policeman searched the storekeeper and the van*, or *the policeman searched the storekeeper or the van*. This is in line with our findings from the previous experiments, which seemed to suggest that comprehenders do not always manage to interpret the garden-path sentences faithfully to the input. The results from open-ended questions also show that the spectrum of incomplete or unfaithful representations can be relatively diverse.

Unlike in the previous experiments, where the question targeting the initial misanalysis led to the lowest response accuracy, the question targeting the dative (possessive) interpretation of the ambiguous noun (*Whose was the van?*) generated most incorrect responses in Experiment 4. The relatively low response accuracy can be explained by the change in the form and target of the question. While in Experiments 1 and 2 the question was very trivial, directly mentioned both the possessor and the possessed (*Did the storekeeper have a van?*) and targeted the possessive relationship itself, in Experiment 4 the question only mentioned the possessed and targeted the possessor. Even though the general response accuracy was low, we did not find a significant effect of sentence type, i.e., the participants did not make more mistakes in the garden-path condition (which we would expect if they struggled with interpreting the ambiguous noun correctly).

We also found a significant effect of animacy for both questions. Sentences with animate objects produced more incorrect answers than sentences with inanimate objects. In the case of QWHAT, the question targeting the initial misanalysis, the presence of animate patients caused the participants to make more mistakes of the substitution type (which is not surprising, since the ambiguous and the disambiguating noun were semantically more similar and thus more interchangeable, causing problems when the participants needed to recall the object). The situation was different for QWHOSE, the question targeting the possessive interpretation of the ambiguous noun, where the errors of the substitution type were stable across all four conditions. Here, the participants produced more "I don't know" responses when the object was

animate. This effect was even stronger for garden-path sentences suggesting that participants might not have been successful in discovering the correct interpretation of the ambiguous noun in some cases, perhaps because they could not use plausibility as a cue during reanalysis, thus being less certain about who the possessor is when compared to the unambiguous condition.

## Experiment 5

Experiment 5 is a replication of Experiment 4 using the word-by-word presentation mode, which allows us to investigate RTs on specific regions of interest. The results can also be compared to Experiment 3, which used the same paradigm but with yes–no comprehension questions. The experiment was preregistered on OSF: https://osf.io/hcf28.

### Hypotheses

The hypotheses related to comprehension accuracy are the same as for Experiment 4 and the hypotheses related to the differences in reaction times are identical with Experiment 3.

### Method

**Participants.**   One hundred forty one Charles University undergraduate students (122 female, 19 male; mean age = 22.82 years) participated in Experiment 5. All participants were native speakers of Czech and participated for course credit.

**Materials.**   The same materials were used as in Experiment 4.

**Procedure.**   In this experiment, word-by-word self-paced reading was employed. Each sentence was thus presented as a series of underscores, and each spacebar press revealed just one word (and simultaneously hid the previous one). Otherwise, the procedure was identical to Experiment 4.

**Data coding.**   An identical data coding procedure was used as in Experiment 4.

**Data analysis.**   Identical analytical steps were followed as in the previous experiments. No participant scored below 75% of correct answers for filler items and thus no one was excluded from the analysis based on this criterion. Data trimming was done the same way as in Experiment 3. First, reaction times below 150 ms were excluded from the analysis. The remaining reaction times were log-transformed and the upper boundary was set as 2.5 standard deviations from the mean, i.e. 7.38 log(ms) or 1600.2 ms. In sum, 2.65% of the data points was excluded. The same contrast coding was used for the linear mixed-effects modeling as in previous experiments. Reaction times were analyzed for separate regions. We were interested in the disambiguating region (i.e. the real object) and the two following regions (to assess potential spillover effects). Random slopes for each region were determined following the approach by Bates et al. [35].

The analysis of response accuracy and the qualitative analysis of the incorrect answers were identical as in Experiment 4.

### Results

**Reaction times.**   Fig 5 shows RTs in ms for each sentence region used in Experiment 5. The analysis targeted the disambiguating region 7 (i.e. the real object) and the two following regions (i.e. 8 and 9). For region 7, sentence type was used as a random slope for participants and object animacy was used as a random slope for items. For regions 8 and 9, sentence type was used as a random slope for participants together with object animacy and for items, sentence type was used as a sole random slope. For region 7, we found separate significant effects

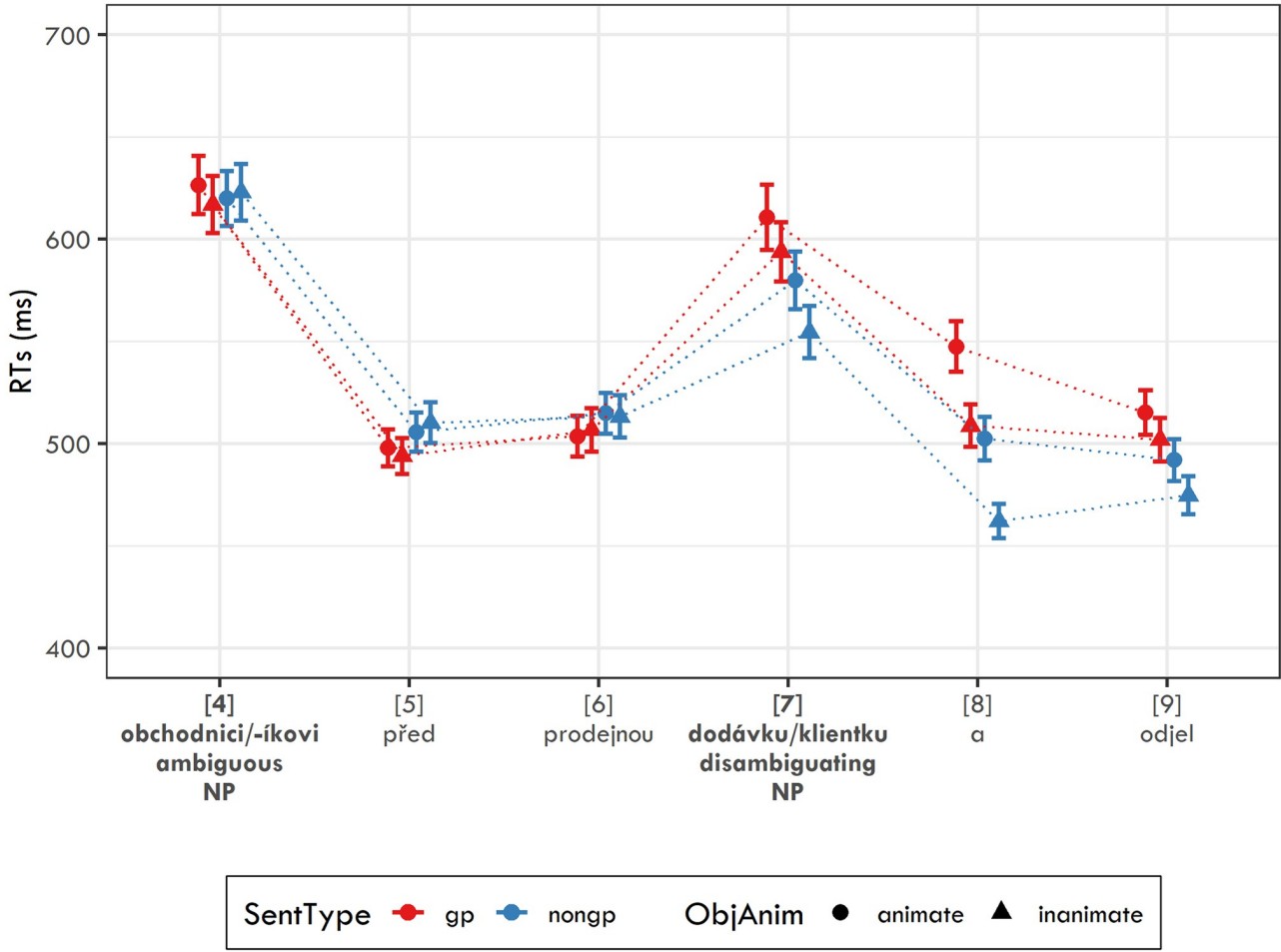

**Fig 5. Reaction times in ms for experimental sentences used in Experiment 5 together with their 95% confidence intervals.**

for sentence type ($\beta$ = 0.053, SE = 0.01, t = 5.162, p < 0.001) and object animacy ($\beta$ = 0.032, SE = 0.013, t = 2.54, p < 0.05), but no interaction effect. For region 8, the model yielded a similar pattern of results with two independent significant effects: sentence type ($\beta$ = 0.077, SE = 0.011, t = 7.024, p < 0.001), object animacy ($\beta$ = 0.065, SE = 0.008, t = 7.894, p < 0.001). For region 9, two independent significant effects were also found: sentence type ($\beta$ = 0.043, SE = 0.008, t = 5.425, p < 0.001), object animacy ($\beta$ = 0.028, SE = 0.007, t = 3.702, p < 0.001).

**Response accuracy.** The overall response accuracy for participants was similar to Experiment 5. The participants' accuracy scores ranged from 33.3% to 100%, the average accuracy being 76.26% (median 79.17%). The between-item variability was 48.57% to 87.86% (mean 76.36%, median 76.95%).

Table 6 and Fig 6 show the response accuracy for each condition. The logit mixed-effects model revealed several significant effects. There was a significant effect of question type ($\beta$ = 0.353, SE = 0.117, z = 3.015, p < 0.01). For questions targeting the initial misanalysis, the model yielded significant effects of sentence type ($\beta$ = −0.796, SE = 0.096, z = −8.311, p < 0.001) and object animacy ($\beta$ = −0.738, SE = 0.096, z = −7.713, p < 0.001). And for questions targeting the possessive relationship, we found a sole significant effect of object animacy ($\beta$ = −0.754, SE = 0.088, z = −8.607, p < 0.001).

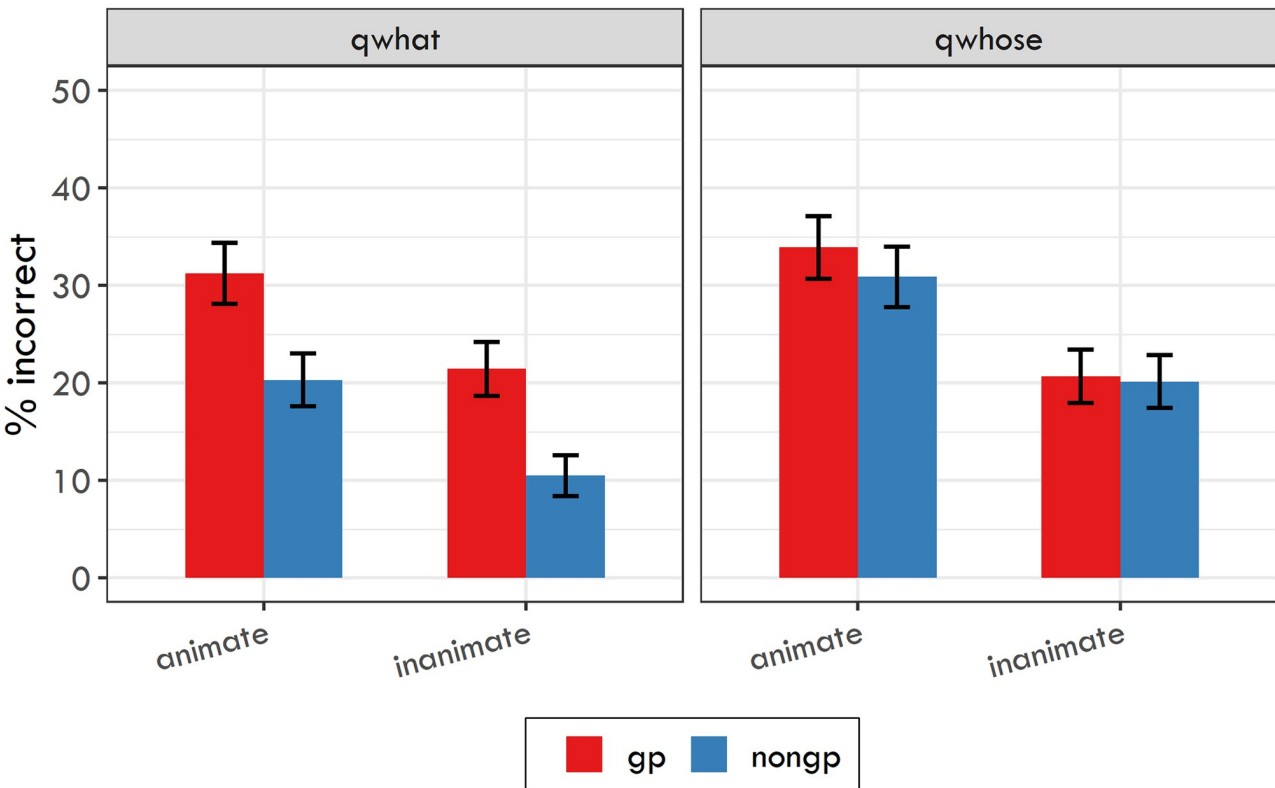

**Fig 6. Response accuracy for the two question types used in Experiment 5 with 95% confidence intervals.** QMIS = question targeting the initial misanalysis; QPOS = question targeting the possessive relationship.

**Incorrect responses.** The frequency of the four different types of incorrect answers is shown in Table 9. The distribution of the incorrect responses was similar to the previous experiment. The "Different" category occurred only twice. The rest of the categories occurred more frequently. For the question targeting the initial misanalysis, most responses belonged to

**Table 9. Frequency of the four types of incorrect responses in Experiment 5.**

| SentType | ObjAnim | QuesType | Error type | | | |
|---|---|---|---|---|---|---|
| | | | I don't know | Different | Substitution | Other |
| gp | anim | qwhat | 16 (6.08%) | 0 | 141 (53.61%) | 106 (40.3%) |
| gp | anim | qwhose | 158 (55.24%) | 1 (0.35%) | 102 (35.66%) | 25 (8.74%) |
| gp | inanim | qwhat | 12 (6.63%) | 0 | 88 (48.62%) | 81 (44.75%) |
| gp | inanim | qwhose | 69 (39.66%) | 0 | 81 (46.55%) | 24 (13.79%) |
| nongp | anim | qwhat | 18 (10.53%) | 1 (0.58%) | 58 (33.92%) | 94 (54.97%) |
| nongp | anim | qwhose | 128 (49.23%) | 0 | 109 (41.92%) | 23 (8.85%) |
| nongp | inanim | qwhat | 12 (13.64%) | 0 | 30 (34.09%) | 46 (52.27%) |
| nongp | inanim | qwhose | 65 (38.24%) | 0 | 92 (54.12%) | 13 (7.65%) |

SentType: *gp* = garden-path sentence; *nongp* = non-garden-path sentence.

ObjAnim: *anim* = animate object; *inanim* = inanimate object.

QuesType: *qwhat* = question targeting the initial misanalysis; *qwhose* = question targeting the possessive relationship.

the "Substitution" and "Other" category. For the question targeting the possessive interpretation of the ambiguous noun, the "I don't know" and "Substitution" categories were the most represented. This pattern was similar for all four conditions.

Further analysis of the Substitution category showed similar results as in Experiment 4. The number of responses that were consistent with the initial misanalysis of garden-path sentences is captured in Table 8.

Once again, garden-path sentences led to visibly more responses consistent with the potential initial misanalysis than non-garden-path sentences. Just as in Experiment 4, the GP type of error was responsible for more than 50% of the responses within the Substitution category for both the high- and low-interference garden-path sentences.

## Discussion

Once again, we found evidence of the participants being garden-pathed. We detected longer RTs on the disambiguating region in garden-path sentences and spillover effects on the two following regions. As predicted, there was also a similar effect of interference on these regions. Similarly to the results of Experiment 3, there was no interaction between ambiguity and interference for RTs on these regions. The results thus suggest that the object animacy does not influence the processing of garden-path sentences directly during reanalysis. If participants rely on information about animacy when interpreting garden-path sentences, they might do so later in the process—for example, while answering the comprehension questions.

The word-by-word presentation mode did not influence response accuracy in a significant way, the results were similar to Experiment 4, where sentences were presented as a whole.

We found a significant effect of sentence type for the question targeting the object of the sentence (QWHAT). The response accuracy was lower when the question followed garden-path sentences. Once again, this difference was caused by mistakes belonging to the substitution category, while the number of mistakes in the remaining categories remained stable across the two conditions. These results strongly support the lingering misanalysis hypothesis. Just as in the previous experiment, we found responses suggesting that participants are sometimes not entirely sure about the interpretation of the sentence or that the representation they formed is not syntactically licit (e.g., *the policeman searched the storekeeper and/or the van*).

The question targeting the possessor once again generated more incorrect responses than the question targeting the object. Although the overall response accuracy was relatively low, we did not detect a significant difference between garden-path and non-garden-path sentences in the number or type of incorrect responses.

There was also an effect of animacy for both questions. For the question targeting the object, the presence of animate patients led the participants to make more mistakes of the substitution type when the sentence had a garden path, which is in line with the assumption that participants may rely on plausibility when deciding between the two possible interpretations of GP sentences, and when the interpretations are semantically similar, they are less successful in interpreting the sentences faithfully. For the non-garden-path sentences, the difference was caused by mistakes belonging to the "other" category, i.e. mostly incomplete and semantically underspecified responses. For the question targeting the possessor, the sentences with animate patients led to more responses belonging to the "I don't know" category. Unlike in Experiment 4, we did not find any interaction between interference and ambiguity in Experiment 5 on general response accuracy. However, the experiments differed in the distribution of the types of incorrect responses, as mentioned earlier.

## General discussion

In this paper, we aimed to examine various aspects of the resulting representation of garden-path sentences in detail. We tested whether comprehenders often create full and faithful syntactic representations of garden-path sentences, even though they fail to delete the initial misanalysis from memory, as assumed by Slattery et el. [21]; or whether resulting representations tend to be more sparse. For example, speakers may form several locally licit syntactic representations [23] or not form a coherent representation of the sentence at all and rely on general inferences instead [20].

To do that, we examined Czech garden-path sentences which contained a local dative-accusative ambiguity of a feminine noun following a transitive verb (e.g., *Ostražitý policista prohledal obchodnici před prodejnou dodávku/klientku a odjel na služebnu.*). We conducted four experiments which differed (a) in their mode of presentation (two used the classic moving-window, word-by-word self-paced reading presentation, two presented the sentences as a whole at once, which allowed the participants to read naturally), and (b) in the type of comprehension questions asked (two experiments used yes–no questions, and two used open-ended, free recall questions). By doing this, we were able to probe whether participants managed to perform all steps that were necessary for creating a faithful and coherent representation of the sentence. If what Slattery et al. [21] assume is correct, participants should be able to correctly attach the disambiguating noun to the syntactic structure in most cases, and they should also be able to interpret the ambiguous noun as dative, i.e., benefactive or external possessor. They should not, however, inhibit the initial misanalysis in their memory.

Our results are largely consistent with the accounts offered both by Slattery and colleagues [21] and by Christianson and colleagues [11]. We found strong evidence suggesting that the initial misanalysis was still active even after participants finished reading the sentence. Thus, we replicated the results of Christianson et al. [11] and various further studies targeting the lingering initial misanalysis. Our results also showed that participants often managed to interpret the rest of the sentence faithfully to the input (they successfully attached the disambiguating noun to the structure and they also reanalyzed the ambiguous noun). However, the evidence of the participants forming a complete and coherent representation was mixed.

In Experiments 2 and 3, which employed yes–no questions, we documented a clear difference in response accuracy for questions targeting the real object between garden-path and non-garden-path sentences. Participants thus often did not form a full and faithful representation after reading garden-path sentences. Needless to say, response accuracy for questions targeting the correct interpretation of the real object was significantly higher than for questions targeting the initial misanalysis. This points out that the problem with forming a full and faithful representation (in the sense of attaching the real object to the transitive verb) occurs to a considerably lower degree than the problem with erasing the initial misanalysis from memory.

Experiments 3 and 4, which employed open-ended questions targeting (a) the grammatically correct and faithful interpretation of the patient and (b) the possessive relationship between the ambiguous noun and the object noun, allowed us a deeper look into the aspects of the resulting representation of garden-path sentences. From a quantitative perspective, we found a clear difference for questions targeting the correct interpretation of the sentence. Participants made more errors while answering these questions after reading garden-path sentences than after reading control sentences. Moreover, the majority of the incorrect responses for garden-path sentences pointed towards lingering misanalysis. However, the qualitative analysis showed that part of the incorrect responses for garden-path sentences occurred due to a failure in forming a complete and coherent analysis of the garden-path sentences.

We may thus say that the general pattern of results somehow conforms to the claims by Slattery et al. [21]. However, our results also bring evidence that the processing of garden-path sentences may often lead to different ends than to forming a full and faithful interpretation of the sentence input.

The findings from the first two experiments show that when participants reach the disambiguating noun, they may not incorporate it into the structure they have built so far and instead choose to stick with the initial misanalysis of the first part of the sentence, while not building a coherent representation of the rest of the clause. This does not seem to happen too often (in comparison to the lingering misanalysis), but it is still well attested in the data.

In other cases, comprehenders may also form a "merged" representation (where both the ambiguous and the disambiguating noun are interpreted as a patient) or they may remain unsure about the meaning of the sentence and choose not to commit to a specific interpretation. In other words, when deciding between the two possible interpretations (*the policeman searched the storekeeper* vs. *the policeman searched the van*), participants sometimes either accept both, or neither. We found multiple responses where the participants directly stated both the ambiguous and the disambiguating noun as the object of the verb (e.g., *prohledal obchodnici a dodávku* 'he searched the storekeeper and the van') and where they directly stated they were not sure about the correct interpretation (e.g., *prohledal obchodnici nebo dodávku* 'he searched the storekeeper or the van').

Slattery and colleagues [21] do not claim that reanalysis of GP sentences is always successful. However, they decide to explore the cases where the correct syntactic structure is built during reanalysis. As was mentioned previously, our results are largely consistent with Slattery's account (in many cases, the disambiguating region and the ambiguous region are interpreted correctly, while the initial misanalysis of the ambiguous region still persists). Nevertheless, the number of trials, where the reanalysis failed and a proper syntactic structure was not built, is significant in our results, and these cases occur reliably and repeatedly across all experiments, regardless of their design. The results from the yes–no comprehension questions, and especially from the open-ended questions, show that the pattern of responses is very diverse. Representations built during processing of the sentences or when answering comprehension questions can be incoherent or partially disrupted in multiple different ways. The accounts offered by Christianson et al. [11], Slattery et al. [21] or Huang and Ferreira [23] are all consistent with some portion of our data. However, the big picture seems to be much more diverse than that. We thus want to advocate for an approach that would focus on reanalysis processes and their results in all their diversity.

Another interesting finding of our study is related to the semantics of the disambiguating region. Sentences with animate patients generated more incorrect responses regardless of the presence of temporary ambiguity. We also found an interaction of object animacy with ambiguity, showing that sentences with animate objects further complicated the processing of garden path sentences and lowered the chance of correctly answering the comprehension question targeting the initial misanalysis.

We assume that this is mostly an effect of encoding [30] and retrieval [9] interference. When the two nouns are both animate and thus semantically more similar, it makes the sentence more difficult to process and it also makes it more difficult to reconstruct correctly when answering the comprehension questions. The effects of encoding interference are observable in the on-line measures, in the form of elevated reading times on the second noun and the following regions. The effect could also be potentially caused by the fact that animate patients are less acceptable or more difficult to process in general [31], making the processing of the whole sentence more difficult for the participants, and making them less willing to accept animate nouns as patients. However, we saw that unambiguous sentences with animate patients were

still judged as acceptable by the participants (even though they were less acceptable than sentences with inanimate patients), so we can assume that the animacy should not be as problematic as to make the participants unwilling to interpret the noun as a plausible patient (i.e., assume that a policeman was searching a client). The cause of the low response accuracy should not lie in the animacy of the object itself.

It is also possible that the potential cause for the decrease in response accuracy lies in the nature of the possessive relationship, since the possessive relationship between two animate entities (*the storekeeper's client*) is qualitatively different from the relationship between an animate and inanimate entity (*the storekeeper's van*), e.g. in terms of alienability [27]. The fact that the nature of the possession might be crucial for answering the questions is supported by the detected significant effects of animacy for questions targeting the possessive relationship. The most likely explanation for the animacy effect on response accuracy is similarity-based interference, however, retrieval of the correct answer to this question should not be influenced by potential interference of the two nouns, since one of the nouns is not involved in the search at all (the possessum is already mentioned in the question and the participants should thus only search for the possessor). It is possible that it is simply more difficult or less plausible to construe a possessive relationship between two animate entities. However, it is important to mention that unambiguous sentences with animate patients still received relatively good acceptability ratings in our study. We can thus assume that the possessive relationship itself was probably not considered too problematic by the participants, and the detected effect of the object animacy was at least to some extent caused by interference, as assumed.

The fact that participants struggle more with finding a coherent and faithful interpretation of garden-path sentences when the two potential object nouns are semantically closer can be explained at least in two ways. First, it is possible that the similarity between the ambiguous noun and the disambiguating noun leads to encoding interference [30]. And since the higher level of interference makes the processing of the sentence more difficult (as documented by higher RTs and lower acceptability ratings), the cognitive load the comprehenders have to face in such a case is higher, and in order to compensate for it, they have to rely more strongly on heuristics while processing the sentence [40]. It has been pointed out multiple times that one such outcome of heuristic processing is the tendency to not inhibit the initial misanalysis of garden-path sentences (and, potentially, not performing a complete reanalysis of these sentences in general). Thus, higher level of interference might cause the initial misanalysis to linger more often, exactly as we have seen.

Another explanation relates to decision making processes during reanalysis or when answering the comprehension questions. As the responses to open-ended questions showed us, there are cases where comprehenders decide between the two possible interpretations of GP sentences ('the policeman searched the storekeeper' vs. 'the policeman searched the van') and sometimes remain unsure even after reading the whole sentence. The plausibility, frequency and general acceptability of these two options might play a role in their decision making. When these two local interpretations are more similar, it is more difficult for the comprehenders to choose one of them over the other, because they are both equally supported. And thus, the number of instances where comprehenders remain undecided, create merged representations or simply give up on trying to interpret the sentence might be higher. Since we failed to detect a significant interaction of ambiguity and animacy on the disambiguating region or on the following regions, we may assume that animacy of the patient does not directly influence the reanalysis and rather comes into play as a post-processing effect, most likely when answering comprehension questions.

The results of the present study also raise another very interesting question, namely the individual differences in garden-path processing. It has already been shown that

comprehenders differ tremendously in what (if any) re-reading strategies they use during reanalysis of GP sentences [41, 42]. In each of the four self-paced reading experiments, we observed participants with a response accuracy of 100% or nearly perfect, but also those who scored much worse. Moreover, participants differed considerably in the magnitude of garden-path effects measured in the difference in reaction times. Unfortunately, the present study was not designed to examine the individual differences in detail. Each participant received only several examples of each condition which prohibits any meaningful analysis of the individual differences. Nevertheless, individual differences in garden-path processing is a potentially fruitful research topic since it seems that native speakers of a language may differ largely in how they process garden-path sentences.

Before we conclude, we should address three more, rather methodological aspects of the study. The first one is concerned with the different presentation modes. In Experiments 2 and 4, we used the whole-sentence-at-once presentation, whereas in Experiments 3 and 5, the traditional moving-window word-by-word self-paced reading mode was used. We found a slight decrease in response accuracy for the word-by-word presentation mode, which can be attributed to the fact that the participants were not able to re-read the material. The decrease mirrors a similar finding by Huang and Ferreira [23], who compared the results from a word-by-word SPR task with the results from an eye-tracking experiment (which also allows the participants to perform regressions). Nevertheless, despite this decrease in accuracy, the general pattern of results stayed the same independently of the presentation mode. In general, this suggests that the traditional moving-window paradigm is well suited for such recall experiments and can also provide researchers with information about the reading process.

The second methodological aspect which needs to be discussed is the use of comprehension questions in order to examine resulting representations. When answering the comprehension questions, participants are forced to recall information, reconstruct the sentences and settle on a single, explicit interpretation. In some cases, such responses might, to a certain extent, differ from the representation created directly during reading. Comprehension questions cannot decisively show whether the comprehenders create a full and faithful representation when processing sentences. Nevertheless, such questions (immediately presented after reading a sentence) offer us an important means to unveil the resulting representations readers create while reading sentences. They provide us with information which cannot be obtained otherwise. As Ferreira and Yang [43] have recently shown, the study of language comprehension should benefit both from the online measures (such as reaction times) and offline measures (such as comprehension questions).

We have seen that, in general, open-ended questions yielded a much higher rate of incorrect answers. This may be caused by two reasons. First, while answering yes–no questions, participants may guess the answer with a 50% probability of success. Second, yes–no questions are typically somehow leading, because they often have to contain cues to the sentence content. For example, we used questions such as *Did the policemen search the client?*, which practically reactivate the subject, the object and the verb of the sentence. Similarly, the classical study by Christianson et al. [11] and many others targeted the resulting representation of sentences such as *While Anna dressed the baby played in the crib* using questions such as *Did Anna dress the baby?*, which also activated three parts of the targeted sentence. This is especially clear in comparison to open-ended questions such as *What did the policeman do?*, which we used in the current study, or even more general questions such as *What happened?* [44].

The influence of question format is especially clear if we compare the results of yes–no and open-ended questions targeting the possessive relationship. In Experiments 2 and 3, which used yes–no questions such as *Did the storekeeper have a van?*, which contained the true possessor and the true object, and the correct answer was thus "yes", the general response accuracy

was relatively high. However, in Experiments 4 and 5, which used general questions such as *Whose was the van?*, the response accuracy was rather low and in some conditions even very similar to the response accuracy for the questions targeting the initial misanalysis. This clearly shows that the presence of cues and a high probability to correctly guess the answer may skew response accuracy to a considerable extent. This is problematic, especially in relation to questions like *Did the storekeeper have a van?* which are often very trivial and easy to answer correctly without any knowledge of the previous sentence. Such questions are in fact not very informative when used to probe sentence representations, because they allow comprehenders to rely on encyclopedic knowledge and inferences.

One possible problem related to the use of open-ended questions relates to the coding and interpretation of the results. We developed simple coding rules to decide which responses were correct and which were not, and for the latter, what type of error they contain. However, this might be done in different ways. Moreover, the problem is that responses to open-ended questions do not give us all information needed to assess the resulting representation of the given participant, since they are rather concise. For example, if someone responds to the question *What did the policeman do?* by saying *He searched the storekeeper*, it is a clear case of a lingering misanalysis, but it does not inform us about the rest of the representation (it might be the case that the given participant formed an otherwise full and complete representation, but also, this could be a case of an incomplete representation). We believe that using open-ended questions is generally more telling than employing traditional yes–no questions, especially since it enables a deeper, qualitative analysis of the data. However, it also has its disadvantages.

As one of the anonymous reviewers pointed out, another methodological problem of the present research is the relatively small number of examples of each condition for each participant (four in Experiments 2 and 3, and six in Experiments 4 and 5). We are fully aware of possible power problems this might present to the analysis. Nevertheless, we believe this does not present a serious problem for the present findings. Crucially, all four self-paced reading experiments used substantial sample sizes of more than 140 participants each and yielded the same pattern of results. Moreover, the creation of experimental items was already highly constrained due to grammatical factors. As we mention in the section Structure of Czech and the Current Study, the number of verbs that can be plausibly combined with both animate and inanimate patients is limited in Czech, and thus, we had to use each verb and disambiguating noun twice to get the current number of experimental items. To get even more items and thus raise the statistical power, we would also need to repeat the ambiguous nouns since they come from a feminine declension class which contains a relatively closed set of nouns denoting people (i.e., the *růže*-paradigm). Finally and importantly, using a substantially higher number of examples of each condition per participant might lead to unfortunate adaptation effects [45, 46]—in a way, the experiment would then train the participants to process the garden-path sentences used in the course of the experiment, which could skew both the reaction times and response accuracy.

## Conclusion

Overall, the results of our experiments suggest that the resulting representations of garden-path sentences may be relatively diverse. The evidence from open-ended questions shows that the form of the final representation varies considerably. In the majority of the trials, comprehenders seem to be able to interpret the garden-path sentences faithfully to the input and delete the initial misinterpretation from memory. In approximately one third of the trials, they succeed in reanalyzing the sentence and manage to attach the disambiguating noun into the structure and reinterpret the ambiguous noun, but they fail to inhibit the initial

misinterpretation. But there is also a significant, albeit smaller, number of trials where comprehenders only construct several local interpretations of the sentence. Sometimes they attempt to merge these local structures together even though the result is not consistent with the input, sometimes they remain unsure about the final interpretation and never build a coherent global representation of the sentence at all, deciding to maintain both of these unconnected local interpretations instead. And sometimes the reanalysis simply fails, and comprehenders only maintain the initial misinterpretation (with the ambiguous noun interpreted as an object) while giving up on the rest of the sentence, or they give up on trying to interpret the sentence altogether.

## Supporting information

**S1 Table. Model examples of incorrect responses to the question What did the policeman do? in Experiments 4 and 5.**
(PDF)

## Acknowledgments

We would like to express our sincere gratitude to Jakub Sláma and James Brand for their invaluable help in proofreading and style cleaning our paper.

## Author Contributions

**Conceptualization:** Markéta Ceháková, Jan Chromý.

**Data curation:** Markéta Ceháková.

**Formal analysis:** Jan Chromý.

**Investigation:** Markéta Ceháková, Jan Chromý.

**Methodology:** Markéta Ceháková, Jan Chromý.

**Project administration:** Markéta Ceháková.

**Supervision:** Jan Chromý.

**Writing – original draft:** Markéta Ceháková, Jan Chromý.

**Writing – review & editing:** Markéta Ceháková, Jan Chromý.

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
