## [Decision Letter · Decision Letter 0]

13 Mar 2023

PONE-D-23-03278Garden-path sentences and the diversity of their (mis)representationsPLOS ONE

Dear Dr. Ceháková,

Thank you for submitting your manuscript to PLOS ONE. After careful consideration, we feel that it has merit but does not fully meet PLOS ONE’s publication criteria as it currently stands. Therefore, we invite you to submit a revised version of the manuscript that addresses the points raised during the review process. Both reviewers agree that the study you report has been carried out rigorously and is of considerable interest to the field. The reviewers also note a number of non-trivial issues, however, which you should address in your revisions. Reviewer #1, in particular, provides many detailed, constructive suggestions as to how your manuscript and the statistical analysis can be improved. Some of Reviewer #2's comments overlap with those made by Reviewer #1, specifically regarding your end-of-trial comprehension questions and what these might measure. Looking more deeply into the latter issue may have important consequences for your theoretical conclusions. Please also attend carefully to the reviewers' more minor comments. Please submit your revised manuscript by Apr 27 2023 11:59PM. If you will need more time than this to complete your revisions, please reply to this message or contact the journal office at plosone@plos.org. Please include the following items when submitting your revised manuscript:A rebuttal letter that responds to each point raised by the academic editor and reviewer(s). You should upload this letter as a separate file labeled 'Response to Reviewers'.A marked-up copy of your manuscript that highlights changes made to the original version. You should upload this as a separate file labeled 'Revised Manuscript with Track Changes'.An unmarked version of your revised paper without tracked changes. You should upload this as a separate file labeled 'Manuscript'.

We look forward to receiving your revised manuscript.

Kind regards,

Claudia Felser, Ph.D

Academic Editor

PLOS ONE

Journal Requirements:

2. Please provide additional details regarding ethical approval in the body of your manuscript. In the Methods section, please ensure that you have specified the name of the IRB/ethics committee that approved your study.

3. Please provide additional details regarding participant consent. In the Methods section, please ensure that you have specified (1) whether consent was informed and (2) what type you obtained (for instance, written or verbal). If your study included minors, state whether you obtained consent from parents or guardians. If the need for consent was waived by the ethics committee, please include this information.

Reviewers' comments:

Reviewer's Responses to Questions

**Comments to the Author**

1. Is the manuscript technically sound, and do the data support the conclusions?

Reviewer #1: Partly

Reviewer #2: Partly

2. Has the statistical analysis been performed appropriately and rigorously? 

Reviewer #1: Yes

Reviewer #2: Yes

3. Have the authors made all data underlying the findings in their manuscript fully available?

Reviewer #1: Yes

Reviewer #2: Yes

4. Is the manuscript presented in an intelligible fashion and written in standard English?

Reviewer #1: Yes

Reviewer #2: Yes

5. Review Comments to the Author

Reviewer #1: The manuscript ‘Garden-path sentences and the diversity of their (mis)representations’ investigates the processing of temporarily ambiguous sentences. The authors report the results from 4 web-based reading experiments and a web-based grammaticality-judgement study. The key results are that (a) an incorrect initial analysis which is activated during processing of the sentence remains active even after processing of the sentence is complete, and that (b) readers differ considerably with regard to which final interpretation they adopt.

The manuscript is written in a scholarly way. The experiments are based on a solid experimental design, and motivated with reference to existing theoretical accounts of ambiguity-resolution. The key question addressed in the manuscript, i.e. the nature of the final interpretation adopted by readers, has received considerable attention in recent publications (see Fujita & Cunnings, 2020, for a detailed discussion of the issue). In this respect, the manuscript is certainly of interest for the scientific community in this field.

That said, I have a number of concerns about the theoretical background, hypotheses, and interpretation of the results. In particular, while the authors motivate their study as a way to test the theoretical accounts proposed by Christianson et al. (2001) and Slattery at al. (2013) against each other, the experimental results reported in the manuscript can actually be accounted for by either of the two accounts. Further, some aspects of the experimental design, in particular the inclusion of the animacy manipulation, are not sufficiently motivated in the current version of the manuscript. I also have some concerns about particular aspects of the statistical analyses. Finally, the results sections currently include a substantial amount of technical details, which makes the manuscript hard to follow for readers.

I elaborate on all these concerns in more detail below, along with suggestions for how the manuscript may be fixed. In sum, the study strikes me as in principle publishable, but requires substantial corrections and rewriting.

General issues

1. The description of the ambiguity resolution account proposed by Slattery and colleagues strikes me as somewhat inaccurate. The account does not claim that reanalysis is always successful and complete. Slattery and colleagues would certainly acknowledge that the processing of a garden-path sentence (just as the processing of any other kind of sentence with a complex syntactic structure) may occasionally fail. Instead, their key point is that the initial analysis may linger even if reanalysis is successful and complete, because the initially activated, but ultimately incorrect analysis leaves a memory trace. This issue is relevant for the present manuscript because it affects the hypotheses stated on page 11: As far as I can see, the results from the vive experiments are largely consistent with both Christianson et al’s (2001) and Slattery et al’s. (2013) theoretical accounts.

As a side note, with regard to Slattery et al’s. (2013) account, there is also a number of recent studies which are directly relevant, but are currently not mentioned in the manuscript. In particular, I suggest that the authors check Cunnings (2017) and Fujita & Cunnings, 2020. There may well be additional relevant work on this issue that I am not aware of.

2. The rationale behind all experiments is based on the key assumption that the answers to post-sentence comprehension questions reflect a participant’s final interpretation of the sentence. However, while this assumption has also been made in a number of previous studies, is not entirely uncontroversial. Even if reanalysis is successful and complete, the initially-activated incorrect interpretation may leave a memory trace. This memory trace may then influence answers to post-sentence comprehension questions: When a participant gets to a post-sentence comprehension question, they must remember/reconstruct the previously-encountered sentence that the question refers to. This reconstruction process may be affected by the lingering initial interpretation of the sentence. This explanation is obviously directly related to Slattery et al.’s (2013) ambiguity resolution account. I suggest that this issue is taken into account when interpreting effects of ambiguity in post-sentence comprehension questions. In sum, I am not sure whether it is possible to differentiate between the different theoretical accounts of ambiguity resolution on the basis of the current results.

3. It is not entirely clear to me why the manipulation of animacy of the disambiguating noun was included in the experimental design. Both the motivation of the experimental design on page 8 and the hypotheses on page 11 only refer to a possible main effect of animacy (i.e. a general effect of animacy which occurs irrespective of whether the sentence is temporarily ambiguous or not), which is not particularly relevant for the research questions (quote from p. 8: “...leading to a lower response accuracy in both the garden-path and the control condition.”). In this respect, the short, interference-based justification for the inclusion of the animacy manipulation provided on page 8 is not quite sufficient. My suggestion would be to discuss how animacy may potentially interact with the other independent variables, particularly with the ambiguity manipulation. For instance, it is theoretically possible that verbs such as searched generally occur more often with inanimate objects than with animate ones. If this were the case, this would lead to additional difficulty when the disambiguating noun is animate.

4. I am a bit sceptical about the way in which question type (i.e. whether the comprehension question following the experimental sentence targets (a) the correct analysis of disambiguating noun, (b) the deactivation of the initial misanalysis, or (c) the correct analysis of the ambiguous noun) is utilized in the statistical analyses. In the model analyses, the authors treat question type as an independent variable. From a purely technical point of view (i.e. if we only look at the abstract properties of the experimental design), question type does indeed constitute an additional independent variable with three levels. At a content level, however, the three types of comprehension questions refer to qualitatively different aspects of the reanalysis process: The three different question types refer to different segments of the sentence, and thus each address a somewhat different research questions. As a results, the key issue is not so much whether the three levels of question type differ from each other (which is what the current analyses in the manuscript address), but how the effects of ambiguity look like within each level of question type. My suggestion would be to present an alternative analysis which takes this property of the question type manipulation into account: As a first step, 2x2 analyses which include only ambiguity (gp vs. non-gp) and animacy (animate vs. inanimate) could be presented. This is justified given that all three question types are similar in the sense that they all target the final interpretation. Following these 2x2 analyses, you could split up the data by question type and provide additional, separate analyses of the answers to the comprehension questions for each question type, in which you look at the effect of ambiguity (and animacy) separately for ‘qcor’, ‘qmis’, and ‘qpos’ questions.

5. Experiments 3 and 5 are essentially exact replications of Experiments 2 and 4, the only difference being that they rely on word-by-word self-paced reading rather than whole-sentence presentation. In principle, self-paced reading data is certainly of interest here, because it gives us insight into what happens during processing of the ambiguous sentence. However, the current version of the manuscript does not provide sufficient motivation for re-doing the exact same experiments again with self-paced reading. Also, both the motivation for the experiments and the hypotheses currently focus strongly on the final representation after processing of the sentence is complete, and do not refer to on-line measures. I suggest that short paragraphs are added to the sections for the two experiments, in which the authors explain what insights the following self-paced reading experiments can provide above and beyond what we already know from the previous whole-sentence experiment.

6. While the number of participants tested in the experimental studies is more or less acceptable for a web-based study of this kind, the experiment is considerably underpowered with regard to the number of items per condition that each participant encountered during testing: Given that each experiment is based on a 2x2x3 latin-square design and contained 48 experimental items, each participant got to see only 48/12 = 4 items per condition. My advice would be to at least provide a justification for why it was unavoidable to work with such a small number of items per condition.

7. I find the way in which the results are reported not very straightforward to read. As they are, the results sections for the five experiments are far too detailed and contain plenty of rather unimportant or technical information. While such a detailed description is certainly useful for the review process, a lot of content should be removed from the paper and instead be included in the supplementary materials, the repository, or in the R script for the analyses. I provide a few suggestions for how the Results sections can be made shorter and easier to read here, but suggest that the authors also look for additions options to achieve this. First, the model tables at the end occur in-between the references, presumably due to a converting error. Quite a few effects and coefficients reported in the model tables are also fully repeated in the text as well. It is enough to just report the exact coefficients only in tables, and to then discuss what the respective significant effects mean in the text. Second, Table 4 refers to a purely technical aspect of the analysis, and might actually confuse readers who are not very experienced with linear-mixed effects models. The table could be included in the supplementary materials or in the R script, but feels a bit misplaced in the manuscript. Third, with respect to the reading time analyses, while it is certainly justified to conduct the model analyses on transformed scores, inverse-transformed reading times are not very intuitive for the reader. Thus, I suggest that the descriptive Figures 2 and 5 should show either untransformed mean reading times or back-transformed values. Also, given that the first experimental manipulation occurs in Region 4, it does not make much sense to show results for the first three segments (In fact, it would be extremely weird if the conditions already differed at a point where they are still exactly identical). Again, these additional results could be moved to the supplementary materials or the repository.

Line-by-line comments

P. 13, l. 550ff.

The authors state that the random-effects structure of their models included “intercept(s) for participants and items, the by-participants random slope for sentence type and a by-item random slope for sentence type”. However, given that all experimental manipulations included in the design were both within-participants and within items manipulations, the maximal random-effects structure (which should ideally be used in analyses of this kind, as long as the model reaches convergence; see Barr, Levy, Scheepers & Tily, 2013) would contain a number of additional random slopes which are not included in the analyses reported in the paper, such as random slopes by participants and items for the animacy manipulation, or random slopes for the interaction between ambiguity and animacy. I acknowledge that such a maximal model would probably not reach convergence in this particular case, but then the random-effects structure would have to be gradually simplified through a standardized procedure, rather than to just pick a simpler random-effects structure which reaches convergence.

P.13, ll. 554 ff.

The paragraph should include a justification for why these particular Helmert contrasts were chosen for the question type manipulation. Especially the second contrast is quite unusual: This is the difference between qmis-questions versus hypothetical questions which are somewhere in the middle between qcor and qpos questions. I find it hard to imagine how such a hypothetical question may look like. Also, I think quite a few readers will struggle to understand this paragraph.

P.16, ll. 664 ff.

The analysis of reading times for the disambiguating region relies on very minimal data trimming, with only reading times above 15000 ms and below 150 ms being considered outliers. If this were a self-paced reading study conducted in a controlled lab environment, this kind of minimal data trimming would perhaps be acceptable, but the fact that this is a web-based study (which makes it likely that the results contain a substantially larger number of extremely high outliers) calls for a more fine-grained check of whether the results are potentially contaminated by the influence of a small number of extremely high values, particularly given that LMERs are quite vulnerable with regard to the effect of such outliers. Also, if it really takes a participant 14999 ms to read a single word, there is definitely something highly unusual going on, so we should conclude that this data point does not reflect natural reading. My advice would be to apply standard cut-off criteria reported in the literature (e.g. exclude all data point which are more than 2 or 2.5 SD away from the overall region mean), and to check whether the key effects remain the same irrespective of which criterion for data trimming is applied.

P. 33, Tables S1 and S2, and P. 34, Tables S3 and S4

These tables are mixed with the list of references. This is probably just a minor copy+paste or .pdf conversion error though.

Reviewer #2: The paper investigates the nature of representations built when comprehending garden-path (GP) sentences. This is done in a series of five experiments with varied sentence presentation mode and question types, allowing for quantitative as well as qualitative analysis of response types. The authors come to a conclusion that the resulting representations are mixed: i.e., sometimes they contain an accurate representation along with the initial misanalysis, sometimes they combine multiple locally licensed representations, and in some cases they only encompass the initial misanalysis.

In my opinion, this is rigorous work that has many strengths:

- The research goals are worthwhile. The qualitative analysis of GP sentence representations is a valuable contribution that brings GP research closer to understanding ‘real-world’ scenarios of comprehending grammatically complex input.

- The methodology is rigorous. I appreciated the design of the project, cleverly manipulating methodologies across experiments. I particularly liked the use of open-ended questions to probe sentence representations. It is nice to see new language material and an understudied language in GP research. The authors used open science practices: the experiments have been pre-registered, the data are publicly available. Sample sizes in all of the experiments are impressive.

- The paper is well-written, well-structured and coherent.

However, I also have several concerns and suggestions, please see below.

Major points

1) For almost all of the results (as well as earlier results by Chromy, 2022), I have been wondering whether they can be due to offline (post-processing) effects. That is, readers may arrive at a coherent representation of the sentence – but then, faced with a comprehension question, they experience a cognitive overload and fail to retrieve correct information from their working memory. In other words, the readers do form a faithful and coherent representation but then, being overloaded with this challenging task followed by the need to also process the question, cannot stand the cognitive load.

The post-processing interpretation seems particularly likely to me in light of interference effects (across experiments). The authors suggest that encoding is more difficult in the high-interference condition. But perhaps the interference effect emerges at a stage later than encoding – I would consider interference in working memory at the retrieval stage, post sentence processing.

I would recommend the authors to consider post-processing effects throughout the paper.

2) I have several concerns about the choice of comprehension questions. I suggest to highlight these as limitations, if my understanding is correct.

- It seems that all comprehension questions targeted the critical (GP) region of the sentence. There were no filler questions targeting other regions of the sentence. That means that when a participant encounters a grammatical GP sentence (without missing or redundant constituents, as in fillers), they can strategize to only focus on the GP region and ignore the rest of the sentence content. From my point of view, this is an important issue that undermines the ecological validity of GP sentence processing in the study.

- Qpos questions (e.g., “Did the storekeeper have a van/a client”) seem rather trivial. I think that these can be answered using common sense and I have difficulty imagining reasons to make an error there. This issue is lightly touched upon in Discussion but I suggest highlighting it more.

3) Statistical models across experiments differ in what random slopes they include, and the selection is not intuitive to me. For example, in Experiment 1, the models include by-participants random slope for sentence type and a by-item random slope for animacy. What motivated the choice of specifically this combination? I suggest using a uniform approach to including random slopes across experiments and explaining it in the paper.

4) Error types are only reported in descriptive terms, without any inferential statistics. Would any inferential statistics on error types across experimental conditions be possible? If generalized linear mixed-effect models or logistic regressions are not feasible due to low numbers of errors, perhaps the authors may consider something like chi-square tests on the ratios of most critical response types across conditions.

Minor points

1) I am not quite clear on how Experiment 1 fits in the global logic of the study. I was particularly confused by the following reasoning:

“We can also see that while garden-path sentences (especially in the high-interference condition) are rated as less acceptable, the unambiguous versions score significantly better, suggesting that potential problems with processing should be indeed attributed to the manipulated variables and not to the general unacceptability of given structures.”

The paper might benefit from a better explanation of what non-trivial results were expected from Experiment 1.

2) All reasoning in the paper seems to rest on the assumption that the readers always follow the garden path and then either recover or do not. In other words, the paper interprets the comprehenders’ behavior as different patterns of recovery from being garden-pathed (with varied success). Can’t there be cases when the readers do not follow the garden path and parse the sentence correctly in the first place? I suggest considering this possibility throughout the discussion of all findings.

Typos

- I noticed some “Exp” instead of “Experiment” (e.g., p. 25).

- Question abbreviations (qpos, qwhose, etc.) might look better in text if highlighted with italics or capital letters.

6. PLOS authors have the option to publish the peer review history of their article (what does this mean?). If published, this will include your full peer review and any attached files.

Reviewer #1: No

Reviewer #2: No

---

## [Author Response · Author response to Decision Letter 0]

26 Apr 2023

Comments

Reviewer #1: 

The manuscript is written in a scholarly way. The experiments are based on a solid experimental design, and motivated with reference to existing theoretical accounts of ambiguity-resolution. The key question addressed in the manuscript, i.e. the nature of the final interpretation adopted by readers, has received considerable attention in recent publications (see Fujita & Cunnings, 2020, for a detailed discussion of the issue). In this respect, the manuscript is certainly of interest for the scientific community in this field.

That said, I have a number of concerns about the theoretical background, hypotheses, and interpretation of the results. In particular, while the authors motivate their study as a way to test the theoretical accounts proposed by Christianson et al. (2001) and Slattery at al. (2013) against each other, the experimental results reported in the manuscript can actually be accounted for by either of the two accounts. Further, some aspects of the experimental design, in particular the inclusion of the animacy manipulation, are not sufficiently motivated in the current version of the manuscript. I also have some concerns about particular aspects of the statistical analyses. Finally, the results sections currently include a substantial amount of technical details, which makes the manuscript hard to follow for readers.

I elaborate on all these concerns in more detail below, along with suggestions for how the manuscript may be fixed. In sum, the study strikes me as in principle publishable, but requires substantial corrections and rewriting.

*** We thank the reviewer for their positive feedback, we revised the paper to address all the critical comments. (see below in more detail).

General issues

1. The description of the ambiguity resolution account proposed by Slattery and colleagues strikes me as somewhat inaccurate. The account does not claim that reanalysis is always successful and complete. Slattery and colleagues would certainly acknowledge that the processing of a garden-path sentence (just as the processing of any other kind of sentence with a complex syntactic structure) may occasionally fail. Instead, their key point is that the initial analysis may linger even if reanalysis is successful and complete, because the initially activated, but ultimately incorrect analysis leaves a memory trace. This issue is relevant for the present manuscript because it affects the hypotheses stated on page 11: As far as I can see, the results from the vive experiments are largely consistent with both Christianson et al’s (2001) and Slattery et al’s. (2013) theoretical accounts.

As a side note, with regard to Slattery et al’s. (2013) account, there is also a number of recent studies which are directly relevant, but are currently not mentioned in the manuscript. In particular, I suggest that the authors check Cunnings (2017) and Fujita & Cunnings, 2020. There may well be additional relevant work on this issue that I am not aware of.

*** We thank the reviewer for pointing us towards the two studies. We included them in the Introduction section in the revised manuscript. We also clarified our interpretation of Slattery et al. (2013) throughout the text. We entirely agree with the reviewer’s interpretation of the account and are grateful for being made aware of the lack of clarity. We do not mean to claim that Slattery and colleagues propose that processing of garden-path sentences never fails, or to test their account against the one made by Christianson and colleagues. What we mean to say is that even when the initial misanalysis lingers, there can be multiple explanations for this phenomenon, and that, in fact, the accounts of Christianson et al. (2001) and Slattery et al. (2013) might both be valid at the same time, because they both represent different (and significant) parts of a wide spectrum of options we can find in the comprehenders’ behavior. Our main aim is to point out that we should focus on the whole range of processing outcomes and not only on a specific, albeit frequent one. We emphasized that our results are consistent with both accounts in the General Discussion section. 

2. The rationale behind all experiments is based on the key assumption that the answers to post-sentence comprehension questions reflect a participant’s final interpretation of the sentence. However, while this assumption has also been made in a number of previous studies, is not entirely uncontroversial. Even if reanalysis is successful and complete, the initially-activated incorrect interpretation may leave a memory trace. This memory trace may then influence answers to post-sentence comprehension questions: When a participant gets to a post-sentence comprehension question, they must remember/reconstruct the previously-encountered sentence that the question refers to. This reconstruction process may be affected by the lingering initial interpretation of the sentence. This explanation is obviously directly related to Slattery et al.’s (2013) ambiguity resolution account. I suggest that this issue is taken into account when interpreting effects of ambiguity in post-sentence comprehension questions. In sum, I am not sure whether it is possible to differentiate between the different theoretical accounts of ambiguity resolution on the basis of the current results.

*** This is an excellent point. We entirely agree that the use of comprehension questions to test final representations of sentences is problematic because the questions target offline mechanisms and the responses may be influenced by numerous post-processing effects. We newly discuss this issue in the General Discussion. Nevertheless, we believe that use of comprehension questions has its advantages and it can offer important insight into the results of processing mechanisms, which would be impossible to obtain with online measures only (for a detailed discussion see Ferreira & Yang, 2019). 

We would also like to thank the reviewer for offering an explanation of how the account offered by Slattery et al. may relate to the post-processing effects. The results may indeed be affected by the memory trace the initial interpretation leaves behind. This explanation is especially consistent with the responses that show the effect of lingering misanalysis – namely positive responses to yes-no questions directed at the initial misanalysis (Did the policeman search the storekeeper?) and responses to open-ended questions (e.g., What did the policeman do?) that explicitly mention the storekeeper as the direct object or mention a level of uncertainty about the correct response (e.g., the policeman searched the storekeeper or the van). However, we also detected a high number of incorrect responses to the question Did the policeman search the van? (i.e., the correct interpretation of the disambiguating region). If this would be a result of post-processing, there would be no reason for the initial misanalysis to influence the response to this question, since the two interpretations might easily co-exist (the scenario where the policeman searches the storekeeper is entirely consistent with the scenario where he searches the van). The only factor blocking this conjoined interpretation is a syntactic one since the verb cannot have two direct objects. This directly contradicts what Slattery et al. predict (i.e., that comprehenders manage to build a proper syntactic representation of the sentence and thus detach the ambiguous noun from the object position). 

3. It is not entirely clear to me why the manipulation of animacy of the disambiguating noun was included in the experimental design. Both the motivation of the experimental design on page 8 and the hypotheses on page 11 only refer to a possible main effect of animacy (i.e. a general effect of animacy which occurs irrespective of whether the sentence is temporarily ambiguous or not), which is not particularly relevant for the research questions (quote from p. 8: “...leading to a lower response accuracy in both the garden-path and the control condition.”). In this respect, the short, interference-based justification for the inclusion of the animacy manipulation provided on page 8 is not quite sufficient. My suggestion would be to discuss how animacy may potentially interact with the other independent variables, particularly with the ambiguity manipulation. For instance, it is theoretically possible that verbs such as searched generally occur more often with inanimate objects than with animate ones. If this were the case, this would lead to additional difficulty when the disambiguating noun is animate.

*** We would like to thank the reviewer for pointing this out. We noticed a difference in response accuracy for animate and inanimate disambiguating nouns in our pilot experiment and decided to control this variable, since it had an effect on the results. We decided to take advantage of this fact, because it let us create a more diverse set of experimental stimuli. Since our research question is directly linked to an assumption that resulting representations of garden path sentences can differ significantly based on numerous factors, we consider having this variety in our experimental design beneficial. However, we completely agree with the reviewer that our motivation for including this manipulation was introduced in a very general manner. We addressed this problem and explained our motivations for including the animacy manipulation in more detail in the Current study section and in the introduction to Experiment 2. We would also like to thank the reviewer for pointing us towards the potential differences in acceptability or frequency of animate and inanimate nouns in the position of the direct object. This is a very good point and we took it into account when discussing what role animacy can play in retrieval or decision making during answering the comprehension questions (as has also been suggested by Reviewer #2). 

4. I am a bit sceptical about the way in which question type (i.e. whether the comprehension question following the experimental sentence targets (a) the correct analysis of disambiguating noun, (b) the deactivation of the initial misanalysis, or (c) the correct analysis of the ambiguous noun) is utilized in the statistical analyses. In the model analyses, the authors treat question type as an independent variable. From a purely technical point of view (i.e. if we only look at the abstract properties of the experimental design), question type does indeed constitute an additional independent variable with three levels. At a content level, however, the three types of comprehension questions refer to qualitatively different aspects of the reanalysis process: The three different question types refer to different segments of the sentence, and thus each address a somewhat different research questions. As a results, the key issue is not so much whether the three levels of question type differ from each other (which is what the current analyses in the manuscript address), but how the effects of ambiguity look like within each level of question type. My suggestion would be to present an alternative analysis which takes this property of the question type manipulation into account: As a first step, 2x2 analyses which include only ambiguity (gp vs. non-gp) and animacy (animate vs. inanimate) could be presented. This is justified given that all three question types are similar in the sense that they all target the final interpretation. Following these 2x2 analyses, you could split up the data by question type and provide additional, separate analyses of the answers to the comprehension questions for each question type, in which you look at the effect of ambiguity (and animacy) separately for ‘qcor’, ‘qmis’, and ‘qpos’ questions.

*** We completely agree with the reviewer that the analysis should focus primarily on the effects of ambiguity within each level of question type. In fact, this is precisely what we did in our analyses. In Experiments 2, 3, 4, and 5, we used nested analysis in the response accuracy logit mixed-effects models. In other words, the ambiguity and animacy were effects nested within the question effect. Thus, the models yielded us results both on the general difference between the question types and on the effects of ambiguity and animacy (in interaction) within each question type. Nevertheless, the model structure was explicitly described only in the Data analysis section of Experiment 2 while for other experiments, we only stated that the analysis was identical as the previous one. To avoid further misunderstanding in the revised version, we now explicitly mention the model structure in the Data analysis section of each experiment. 

5. Experiments 3 and 5 are essentially exact replications of Experiments 2 and 4, the only difference being that they rely on word-by-word self-paced reading rather than whole-sentence presentation. In principle, self-paced reading data is certainly of interest here, because it gives us insight into what happens during processing of the ambiguous sentence. However, the current version of the manuscript does not provide sufficient motivation for re-doing the exact same experiments again with self-paced reading. Also, both the motivation for the experiments and the hypotheses currently focus strongly on the final representation after processing of the sentence is complete, and do not refer to on-line measures. I suggest that short paragraphs are added to the sections for the two experiments, in which the authors explain what insights the following self-paced reading experiments can provide above and beyond what we already know from the previous whole-sentence experiment.

*** The crucial reason for doing word-by-word self-paced reading was to assess the garden-path effects for the sentences examined using on-line measures and not only off-line measures. We believe this is important, because otherwise we would not be sure whether the garden-path effects are present at all in the sentences (since these are new structures that have not been used in an experiment before). As the reviewer suggests, we now corroborate the discussion of the on-line measures and their additional value in comparison to the whole-sentence experiments. 

6. While the number of participants tested in the experimental studies is more or less acceptable for a web-based study of this kind, the experiment is considerably underpowered with regard to the number of items per condition that each participant encountered during testing: Given that each experiment is based on a 2x2x3 latin-square design and contained 48 experimental items, each participant got to see only 48/12 = 4 items per condition. My advice would be to at least provide a justification for why it was unavoidable to work with such a small number of items per condition.

*** Thanks for pointing this out. There are several reasons for this approach. First, it is in general very hard to create more stimuli which would have the same structure, would be sensible, which would not contain further ambiguities (morphological, semantic…) and which could be well targeted by the types of comprehension questions needed. Second, the fact that we use only few examples of each condition for each participant is not necessarily a negative. We are now working on a study which shows clear learning effects in recall of information over the course of the experiment – it seems clear that the more the participants are asked about certain information, the better they are able to recall it. Moreover, other studies (e.g., Fine et al., 2013) show that participants can get adapted to garden-path sentences during the experiment. From this point of view, the use of a limited number of examples of each condition (with a correspondingly large number of participants) might even be considered an advantage, since the results are not biased by learning (syntactic adaptation) to much extent. Third, we documented the same pattern of results across the four self-paced reading experiments. If the small number of examples of each condition for each participant would bias the results, we should see much more variability in the results. We now added a new paragraph to the General discussion which discusses these issues. 

7. I find the way in which the results are reported not very straightforward to read. As they are, the results sections for the five experiments are far too detailed and contain plenty of rather unimportant or technical information. While such a detailed description is certainly useful for the review process, a lot of content should be removed from the paper and instead be included in the supplementary materials, the repository, or in the R script for the analyses. I provide a few suggestions for how the Results sections can be made shorter and easier to read here, but suggest that the authors also look for additions options to achieve this. First, the model tables at the end occur in-between the references, presumably due to a converting error. Quite a few effects and coefficients reported in the model tables are also fully repeated in the text as well. It is enough to just report the exact coefficients only in tables, and to then discuss what the respective significant effects mean in the text. Second, Table 4 refers to a purely technical aspect of the analysis, and might actually confuse readers who are not very experienced with linear-mixed effects models. The table could be included in the supplementary materials or in the R script, but feels a bit misplaced in the manuscript. Third, with respect to the reading time analyses, while it is certainly justified to conduct the model analyses on transformed scores, inverse-transformed reading times are not very intuitive for the reader. Thus, I suggest that the descriptive Figures 2 and 5 should show either untransformed mean reading times or back-transformed values. Also, given that the first experimental manipulation occurs in Region 4, it does not make much sense to show results for the first three segments (In fact, it would be extremely weird if the conditions already differed at a point where they are still exactly identical). Again, these additional results could be moved to the supplementary materials or the repository.

*** We would like to thank reviewer for their suggestions. In the revised version, we changed the RTs plots. Now, we use RTs in ms on y-axis and we show only the regions which are relevant for the analysis (beginning with region 4 which is the ambiguous NP and ending with region 9 which is the disambiguating NP + 2 region). The tables containing the full models for the response accuracy analyses were already in the Supporting information section in the first version of the manuscript – there was only some problem with the PlosONE LaTeX template which made them appear mixed with the references. We now moved the Supporting Information section to the very end of the paper which prevents the mixing. 

The only thing we would prefer not to change is the Table 4 which specifies the coding scheme for the model. We find the information it contains to be very important for understanding what is being modelled and we thus believe, readers who would focus more on the statistical issues in the paper would rather miss it in the main text if we would move it to Supporting information. Also, this table does not take much space so we hope its presence should not really decrease the readability of the paper.

Line-by-line comments

P. 13, l. 550ff.

The authors state that the random-effects structure of their models included “intercept(s) for participants and items, the by-participants random slope for sentence type and a by-item random slope for sentence type”. However, given that all experimental manipulations included in the design were both within-participants and within items manipulations, the maximal random-effects structure (which should ideally be used in analyses of this kind, as long as the model reaches convergence; see Barr, Levy, Scheepers & Tily, 2013) would contain a number of additional random slopes which are not included in the analyses reported in the paper, such as random slopes by participants and items for the animacy manipulation, or random slopes for the interaction between ambiguity and animacy. I acknowledge that such a maximal model would probably not reach convergence in this particular case, but then the random-effects structure would have to be gradually simplified through a standardized procedure, rather than to just pick a simpler random-effects structure which reaches convergence.

*** There is a controversy in the literature on how to approach the random-effects structure of the mixed-effects models. We are well acquainted with the Barr et al. (2013) approach according to which a maximal random-effects structure should be used. However, the problem of this approach may lie not only in convergence problems, but also in the possible overparametrization of the models which could lead to uninterpretable models which is thoroughly discussed for example by Bates, Kliegl, Vasishth, and Baayen (2015): Parsimonious mixed models. In our analysis, we followed their recommendations which does not necessarily expect maximal models. We explicitly mention this in the revised version of the manuscript. 

P.13, ll. 554 ff.

The paragraph should include a justification for why these particular Helmert contrasts were chosen for the question type manipulation. Especially the second contrast is quite unusual: This is the difference between qmis-questions versus hypothetical questions which are somewhere in the middle between qcor and qpos questions. I find it hard to imagine how such a hypothetical question may look like. Also, I think quite a few readers will struggle to understand this paragraph.

*** The contrast coding was chosen in accordance with the recommendations by Schad et al. (2020): How to capitalize on a priori contrasts in linear (mixed) models: A tutorial. The coding aimed to reflect our hypotheses which were based on the previous literature. We did not expect a difference between qcor and qpos (this was targeted by Contrast 1) and we expected qmis to produce more errors than the other two questions (Contrast 2). The other coding types such as treatment coding or sum contrast coding do not enable us a similar type of analysis. 

P.16, ll. 664 ff.

The analysis of reading times for the disambiguating region relies on very minimal data trimming, with only reading times above 15000 ms and below 150 ms being considered outliers. If this were a self-paced reading study conducted in a controlled lab environment, this kind of minimal data trimming would perhaps be acceptable, but the fact that this is a web-based study (which makes it likely that the results contain a substantially larger number of extremely high outliers) calls for a more fine-grained check of whether the results are potentially contaminated by the influence of a small number of extremely high values, particularly given that LMERs are quite vulnerable with regard to the effect of such outliers. Also, if it really takes a participant 14999 ms to read a single word, there is definitely something highly unusual going on, so we should conclude that this data point does not reflect natural reading. My advice would be to apply standard cut-off criteria reported in the literature (e.g. exclude all data point which are more than 2 or 2.5 SD away from the overall region mean), and to check whether the key effects remain the same irrespective of which criterion for data trimming is applied.

*** We are thankful to the reviewer for noticing this. In data trimming, we followed Baayen and Milin (2010): Analysing reaction times who recommend to exclude only clearly discontinous RTs. Nevertheless, as the reviewer suggests, 15 000 ms seem to be a relatively suspicious cut-off point. We thus decided to check the data trimming again and we found it was not done optimally. After a careful examination of the RTs distribution using q-q plots, we set a new cut-off point to 6500 ms in Exp3 and 5,500 ms in Exp5. Needless to say, this did not change the pattern of results in any visible way and also, the general percentage of data points excluded remained very low (i.e., 0.6% in Exp3 and 0.23% in Exp5).

P. 33, Tables S1 and S2, and P. 34, Tables S3 and S4

These tables are mixed with the list of references. This is probably just a minor copy+paste or .pdf conversion error though.

*** This was due to a technical issue with the PlosONE LaTeX template which we were unable to fix. We now moved the Supporting Information to the end of the document (after bibliography) which solved the problem. 

Reviewer #2: 

Major points

1) For almost all of the results (as well as earlier results by Chromy, 2022), I have been wondering whether they can be due to offline (post-processing) effects. That is, readers may arrive at a coherent representation of the sentence – but then, faced with a comprehension question, they experience a cognitive overload and fail to retrieve correct information from their working memory. In other words, the readers do form a faithful and coherent representation but then, being overloaded with this challenging task followed by the need to also process the question, cannot stand the cognitive load.

The post-processing interpretation seems particularly likely to me in light of interference effects (across experiments). The authors suggest that encoding is more difficult in the high-interference condition. But perhaps the interference effect emerges at a stage later than encoding – I would consider interference in working memory at the retrieval stage, post sentence processing.

I would recommend the authors to consider post-processing effects throughout the paper.

*** We would like to thank the reviewer for this comment. This is a very good and very important point. As we mentioned in the response to Reviewer #1, we discussed this issue in General Discussion and emphasized it as a limitation of our study. We are especially grateful for the suggestion concerning interference effects. We included it in the interpretation of the results and in General Discussion. We believe interference might still play a role directly during the processing, because we detected elevated RTs on the second noun (and on the following regions) in the animate condition, but it seems very probable that it comes into play in the later stages as well. 

2) I have several concerns about the choice of comprehension questions. I suggest to highlight these as limitations, if my understanding is correct.

- It seems that all comprehension questions targeted the critical (GP) region of the sentence. There were no filler questions targeting other regions of the sentence. That means that when a participant encounters a grammatical GP sentence (without missing or redundant constituents, as in fillers), they can strategize to only focus on the GP region and ignore the rest of the sentence content. From my point of view, this is an important issue that undermines the ecological validity of GP sentence processing in the study.

***This is a good point that we will try to take into consideration in future experiments. However, we believe it is not a serious problem in this case. It is true that the comprehension questions targeted the first half of the sentence every time, however, this part of the sentence still contains various types of information, and all of the questions were focused on different parts of the sentence. For example, in the case of the sentence Roztržitý kastelán zamkl průvodkyni na nádvoří koloběžku… („The absent-minded castle manager locked the guide’s scooter in the courtyard...“) the qmis question (Did the castle manager lock the guide?) targeted the ambiguous noun (průvodkyni), the qcor question (Did the castle manager lock the scooter?) targeted the disambiguating noun (koloběžku), and the qpos question targeted the potential possessive relationship. Similarly, the open-ended questions also targeted different types of information, and thus, different regions of the sentence (What did the castle manager do?; Whose was the scooter?). We would also like to mention that the number of trials, where participants encountered a specific combination of a sentence and a question, was rather limited. The participants were presented with 24 garden-path sentences (half of them with animate nouns, half with inanimate nouns) and 3 (or 2) types of questions. Overall, they encountered the same condition only 8 (or 12) times out of 150 trials, if we do not count the difference between the animate and inanimate sentences. 

- Qpos questions (e.g., “Did the storekeeper have a van/a client”) seem rather trivial. I think that these can be answered using common sense and I have difficulty imagining reasons to make an error there. This issue is lightly touched upon in Discussion but I suggest highlighting it more.

*** We agree that this is a limitation and we emphasized it in the discussion of the results of Experiment 4 and in General Discussion. 

3) Statistical models across experiments differ in what random slopes they include, and the selection is not intuitive to me. For example, in Experiment 1, the models include by-participants random slope for sentence type and a by-item random slope for animacy. What motivated the choice of specifically this combination? I suggest using a uniform approach to including random slopes across experiments and explaining it in the paper.

*** Good point. We set the random effects structure following the recommendations given by Bates, Kliegl, Vasishth, and Baayen (2015): Parsimonious mixed models. We mention this in the revised version.

4) Error types are only reported in descriptive terms, without any inferential statistics. Would any inferential statistics on error types across experimental conditions be possible? If generalized linear mixed-effect models or logistic regressions are not feasible due to low numbers of errors, perhaps the authors may consider something like chi-square tests on the ratios of most critical response types across conditions.

*** This is a good question. To be honest, we are relatively restrained from doing inferential statistics on error types. First, as the reviewer rightly points out, there is a problem in using mixed-effects models due to the low number of errors and therefore due to huge singularity issues. Second, the techniques we could reasonably use (such as the chi-square test suggested by the reviewer) may be relatively misleading since they do not account for the differences between items and participants. Moreover, the analysis of the error types was aimed to be rather exploratory and qualitative from the start. Its aim was to show the range of the errors participants made in responding the comprehension questions. In sum, we are not strictly against using inferential statistics here, but we do not find it to be a necessary thing to do to enhance the manuscript or its findings. 

Minor points

1) I am not quite clear on how Experiment 1 fits in the global logic of the study. I was particularly confused by the following reasoning:

“We can also see that while garden-path sentences (especially in the high-interference condition) are rated as less acceptable, the unambiguous versions score significantly better, suggesting that potential problems with processing should be indeed attributed to the manipulated variables and not to the general unacceptability of given structures.”

The paper might benefit from a better explanation of what non-trivial results were expected from Experiment 1.

*** Thank you for pointing this out, we rephrased the introduction of Experiment 1 to reflect our motivations and concerns better. Since the stimuli we used in Experiment 1 do not come from actual usage, but were created for the purpose of this study (and contain a rather specific structure which has not been previously analyzed experimentally), we needed to make sure they would be considered acceptable by the participants (i.e., the unambiguous condition would be relatively easy to interpret). If this was not the case, we could not be sure about what causes the effects on response accuracy – it could simply be a result of the sentences being difficult to interpret in general. 

2) All reasoning in the paper seems to rest on the assumption that the readers always follow the garden path and then either recover or do not. In other words, the paper interprets the comprehenders’ behavior as different patterns of recovery from being garden-pathed (with varied success). Can’t there be cases when the readers do not follow the garden path and parse the sentence correctly in the first place? I suggest considering this possibility throughout the discussion of all findings.

*** This is a very interesting point, thank you. To look into it, we tried to examine the strength of garden-path effects for each participant in Experiment 3 using difference between means for garden-path and control conditions and the corresponding Cohen’s d. What we found was that the effect size varied across the participants (with a group of people whose Cohen’s d was even below 0.2). Interestingly, we also found that in some cases, the effect size was above 0.2, but in the opposite direction than expected (RTs in control sentences were slower than in gp sentences). However, we must take into account that the mean RTs for each participant are based on very small number of data points (12) and their standard deviations are rather large. In other words, the values we calculated seem to be very unstable and could change largely if each participant would get more examples of experimental items in the experiment. We thus think of this as a preliminary finding which should be tested in an experiment designed directly to examine individual differences in garden-path processing. In the revised version of the paper, we included a short paragraph on the possibility of individual differences in GP processing as the reviewer suggested. 

Typos

- I noticed some “Exp” instead of “Experiment” (e.g., p. 25).

*** Thanks for noticing this, corrected.

- Question abbreviations (qpos, qwhose, etc.) might look better in text if highlighted with italics or capital letters.

*** Good idea, we now use small capitals to highlight these abbreviations.

---

## [Decision Letter · Decision Letter 1]

8 Jun 2023

PONE-D-23-03278R1Garden-path sentences and the diversity of their (mis)representationsPLOS ONE

Dear Dr. Ceháková,

Thank you for submitting your manuscript to PLOS ONE. After careful consideration, we feel that it has merit but does not fully meet PLOS ONE’s publication criteria as it currently stands. Therefore, we invite you to submit a revised version of the manuscript that addresses the points raised during the review process.

We look forward to receiving your revised manuscript.

Kind regards,

Claudia Felser, Ph.D

Academic Editor

PLOS ONE

Additional Editor Comments:

Both reviewers acknowledge that you have addressed their previous comments very well, but in their second reviews both have raised a number of additional points which I am asking you to address in a final round of major revisions. The most serious point (raised by Reviewer #1) concerns the way your data was trimmed and what consequences your choices might have for your statistical results. Reviewer #2 also makes several helpful suggestions for further improving your manuscript.

Reviewers' comments:

Reviewer's Responses to Questions

**Comments to the Author**

1. If the authors have adequately addressed your comments raised in a previous round of review and you feel that this manuscript is now acceptable for publication, you may indicate that here to bypass the “Comments to the Author” section, enter your conflict of interest statement in the “Confidential to Editor” section, and submit your "Accept" recommendation.

Reviewer #1: (No Response)

Reviewer #2: (No Response)

2. Is the manuscript technically sound, and do the data support the conclusions?

Reviewer #1: Partly

Reviewer #2: Yes

3. Has the statistical analysis been performed appropriately and rigorously? 

Reviewer #1: No

Reviewer #2: Yes

4. Have the authors made all data underlying the findings in their manuscript fully available?

Reviewer #1: Yes

Reviewer #2: Yes

5. Is the manuscript presented in an intelligible fashion and written in standard English?

Reviewer #1: Yes

Reviewer #2: Yes

6. Review Comments to the Author

Reviewer #1: The manuscript has considerably improved with regard to a number of issues raised in the previous review round. In particular, the authors have included a justification for replicating the two experiments based on whole-sentence presentation (Experiments 2 and 4) with self-paced reading (Experiments 3 and 5). The revised version also includes a clearer discussion of how the findings relate to theoretical accounts of lingering interpretations. Finally, the authors have also made a number of useful revisions to the background section which considerably improve readability and clarity of the manuscript.

However, a crucial issue which definitely has to be addressed before publication of the manuscript can be considered is how data trimming/outlier exclusion was conducted in the statistical analyses, particularly for the critical regions in the two self-paced reading experiments. The analyses currently reported in the manuscript are based on an extremely liberal criterion for outlier exclusion, with extremely long reading times which clearly do not reflect natural reading still included in the analysis. A brief reanalysis for the reading time data from Experiment 3 that I have conducted with the data set provided on OSF suggests that the results may indeed be influenced by a small number of extreme values which have survived data trimming. I elaborate on this issue in considerable detail below, and have included some suggestions for how this issue can be fixed. Another, relatively minor issue which should still be addressed is that, while readability of the manuscript overall has considerably improved, the Results sections for the five experiments are still difficult to follow and contain a multitude of tables and figures.

In sum, I have read the revised version with great interest, and I believe that the five experiments reported in the manuscript potentially constitute a valuable contribution to the field. That said, the issue of outlier exclusion strikes me as particularly important: It obviously makes a crucial difference whether the garden-path effects reported in the manuscript really exist, or whether they are caused by a somewhat larger proportion of extreme values in the garden-path than in the control conditions. The issue therefore has to be addressed in a responsible manner.

Outlier Exclusion

- In the first version of the manuscript, the authors had relied on an extremely liberal method for outlier exclusion, and had excluded only very extreme reading times above 15000ms. Even in the revised version, the cut-offs applied (6500ms for Experiment 3, 5500ms for Experiment 5) are far too liberal for web-based self-paced reading experiments of this kind. Given that the critical disambiguating regions in these experiments consist of a single 6-8 letter word, even a reading time of more than 2 seconds is a clear indication that the measure does not reflect natural reading. For the present study, however, even a reading time of 5000 ms is not excluded from the analysis. The authors justify their approach to outlier exclusion by arguing that they have followed the recommendations from Baayen & Milin’s (2010) tutorial for reaction time analysis. However, applying the approach suggested in this tutorial to the present data set mechanically (i.e. without considering what the measures actually represent in this specific study) strikes me as not quite suitable, for at least two reasons: First, Baayen & Milin’s recommendation to only conduct minimal data trimming and to remove only clearly discontinuous reading times refers to the analysis of reaction times in general, and does not take into account what these reaction time measures represent in a concrete study (such as properties of the stimulus for which the reaction times were measured). Thus, their recommended approach is based exclusively on the distribution of data points, and does not discuss any content-based common-sense criteria for what constitutes an outlier. For the reading times measured in the present study, we know that a healthy, fully literate, adult native speaker of a language should definitely be able to read a single 7-letter word in considerably less than 2 seconds. As a result, an extreme reading time of, for instance, 5000ms should definitely be considered an outlier, irrespective of the distribution of data points. Second, Baayen & Milin’s tutorial refers to experimental reaction time studies conducted in a controlled lab environment. Data sets from such studies typically contain only a relatively small number of outliers because the controlled surroundings ensure that a participant only rarely gets distracted from the task. In the current web-based study, in contrast, the experimenter had no control of the participant’s surroundings. As a result, outliers are a potentially much more severe issue in web-based studies. In sum, I thus strongly recommend that the authors revise their analyses and rely on an established approach to outlier exclusion used in previous self-paced reading studies (e.g. first exclude extreme values which clearly do not reflect natural reading, then also exclude reading times which are more than 2 standard deviations above or below the overall mean reading time for the respective segment).

Note that the influence of outliers is a particularly crucial issue here, particularly given that linear-mixed effects models are known to be fairly vulnerable when it comes to the influence of a small number of extreme data points. I thus conducted a brief analysis for the critical disambiguating segment in Experiment 3 myself, with the data provided by the authors on OSF. The results from this more conservative analysis suggest that the crucial effect of ambiguity reported for the disambiguating segment in Experiment 3 is at least partly driven by the influence of a small number of outliers in the data, and becomes considerably smaller when these outliers are excluded: If I conduct the analysis in the same way as in the manuscript (i.e. exclusion of data points above 6500 ms and below 150 ms), mean reading times by condition for the disambiguating segment (i.e. Segment 7) obviously show the very same data pattern as in Figure 2:

Garden-path, animate: 694 ms (SD: 642)

Non-garden-path, animate: 602 ms (SD: 462)

Garden-path, inanimate: 625 ms (SD: 521)

Non-garden-path, inanimate: 548 ms (SD: 376)

These results show clear numerical trends for a garden-path effect, with a 92 ms difference between garden-path- and non-garden path trials for animate items and 77 ms for inanimate items (Note, however, that the huge standard deviations already suggest a presence of a small number of extreme data points.). However, if data points which are more than 2 standard deviations above or below the overall mean reading time for the segment are excluded (an established procedure for outlier exclusion used in a number of previous reading studies), the respective garden path-effects are very considerably smaller, with only a 27 ms difference for animate and 25 ms for inanimate items:

Garden-path, animate: 558 ms (SD: 303)

Non-garden-path, animate: 531 ms (SD: 272)

Garden-path, inanimate: 534 ms (SD: 275)

Non-garden-path, inanimate: 509 ms (SD: 234)

In sum, all this calls for a through exploration of the potential influence of outliers in the data, for all experiments reported in the manuscript. It is crucial to determine whether the key effects reported in the manuscript really reflect longer average reading times for garden-path- than for non-garden-path sentences, or whether the effect is instead driven by a slightly higher number of outliers in the garden-path conditions.

Additional minor issues:

- In their response letter, the authors say that, based on reviewer suggestions in the previous review round, they have revised Figures 2 and 5 so that they show only segments after the experimental manipulation, and also display untransformed reading times instead of inverse-transformed values. However, these updated graphs are not included in the revised version of the manuscript. I suspect this is just due to some sort of converting error.

- The analyses are still considerably too detailed; the revised manuscript contains far too many tables and graphs. Some of the tables should either be merged, included in the supplementary materials, moved to OSF, or deleted entirely (with the key results instead reported in the text).

- As suggested in the previous review round, the authors have added an additional motivation for re-doing Experiments 2 and 4 with self-paced reading (i.e. Experiments 3 and 5) on page 19 (ll. 752-765). While it strikes me as reasonable to motivate these studies with reference to the possibility to look at garden-path effects separately for each segment, arguing that self-paced reading is interesting because it rules out the possibility to go back to a previous segment of the sentence feels a bit weird to me. This essentially characterizes self-paced reading as an artificial research method which does not allow participants to read the sentences naturally.

- In the paragraph discussing issues of statistical power (particularly the number of items per condition each participant encounters during test session) in the Discussion section on page 39, it strikes me as unnecessary to characterize this as a crucial limitation. My suggestion would be to instead include a brief explanation of why it was not possible or problematic to construct a larger set of materials, for instance to do properties of the language.

Reviewer #2: The authors have carefully revised the manuscript and addressed my concerns very well. I think that the manuscript has much improved, particularly in the interpretation of the evidence. I only have remaining minor comments and suggestions.

Minor points

1) The manuscript has become quite lengthy. Of course, it is up to the editors to decide if that is fine, but my advice is to shorten it to improve readability. These are some examples of places that may be shortened:

- Detailed descriptions of previous studies in the Introduction;

- Motivation of design of Experiment 3 (lines 752-765);

- Perhaps some more of the overlapping information in Methods of Experiments 3-5;

- Discussion of methodological aspects in the Discussion section (lines 1421-1497) – these are important but can be expressed with fewer words.

2) I would like to clarify my minor point #2 from the previous review round. I meant not only individual differences between readers in whether they are susceptible to garden-path effects but also across-trial differences in whether the reader follows the garden path. In other words, I meant that possibly in some trials, the participants did not follow the garden path at all and parsed the sentence correctly in the first place. Perhaps the authors could consider whether any of the findings could be accounted by this processing pattern, rather than by any patterns of recovery from being garden-pathed. However, I do not insist on incorporating this into the manuscript.

3) Phrasing of some of the hypotheses has become somewhat vague:

- Hypothesis of Experiment 1 (line 573-574): “there will not be a big difference any significant differences in response accuracy”;

- Hypothesis of Experiment 4 (line 945): “there should be no (or little) difference in response accuracy”.

I feel that it is quite vague to discuss “big differences” or “little differences” (unless we are discussing effect sizes, which is not the case here), so wording in terms of statistical significance would be more rigorous.

4) I apologize for missing this issue at my first reading but I do not agree with the authors’ description of heuristics with reference to Ferreira & Patson (2007). The authors claim that the heuristic is “the tendency to not inhibit the initial misanalysis of garden-path sentences”. But Ferreira & Parson (2007) use the term ‘heuristics’ to describe the parsing mechanisms per se (for example, late closure heuristic, minimal attachment heuristic, reliance on plausibility, etc.) rather than any mechanisms used to deal with conflicting parses. In other words, the initial misanalysis of garden-path sentences is indeed created based on heuristics but the tendency to not inhibit the initial misanalysis is a separate phenomenon.

5) I suggest some editorial changes:

- Lines 1084-1085: “These results are once 1084 again in line with previously mentioned accounts of garden-path sentence processing” – Please clarify which accounts are meant.

- Line 1322-1323: “Slattery and colleagues [21] do not claim that reanalysis of GP sentences does not fail occasionally” – Double negation is difficult to process, please rephrase.

- Lines 1337-1339: “It has already been shown that comprehenders differ tremendously in what (if any) re-reading strategies they use during reanalysis of GP sentences [42,43]” – This sentence looks stranded in this particular spot. I would consider moving this idea to the new paragraph on individual differences (line 1410 and further).

6) Please proofread the paper for punctuation and possibly typos, for example:

- Line 270: “open ended” – A hyphen is missing.

- Lines 452-454: “as well as the experimental items being created specifically for the purposes of our experiments and did not come from actual usage” – Perhaps “were created” is meant.

- Line 1086: “However, we would like to note, that we also detected” – There is an unnecessary comma.

- Lines 1214-1215: “for example while answering the comprehension questions” – A comma is missing.

- Lines 1274-1275: “(they successfully attached the 1274 disambiguating noun to the structure and they also reanalyzed the ambiguous noun.)” – The full stop should follow the bracket.

- Line 1346-1347: Perhaps there should be no paragraph break here.

- Line 1371: “similarity based interference” - A hyphen is missing.

7. PLOS authors have the option to publish the peer review history of their article (what does this mean?). If published, this will include your full peer review and any attached files.

Reviewer #1: No

Reviewer #2: No

---

## [Author Response · Author response to Decision Letter 1]

27 Jun 2023

Reviewer #1: 

The manuscript has considerably improved with regard to a number of issues raised in the previous review round. In particular, the authors have included a justification for replicating the two experiments based on whole-sentence presentation (Experiments 2 and 4) with self-paced reading (Experiments 3 and 5). The revised version also includes a clearer discussion of how the findings relate to theoretical accounts of lingering interpretations. Finally, the authors have also made a number of useful revisions to the background section which considerably improve readability and clarity of the manuscript.

*** We thank the reviewer for their positive feedback, especially for the note about the improved readability and clarity of the manuscript. 

However, a crucial issue which definitely has to be addressed before publication of the manuscript can be considered is how data trimming/outlier exclusion was conducted in the statistical analyses, particularly for the critical regions in the two self-paced reading experiments. The analyses currently reported in the manuscript are based on an extremely liberal criterion for outlier exclusion, with extremely long reading times which clearly do not reflect natural reading still included in the analysis. A brief reanalysis for the reading time data from Experiment 3 that I have conducted with the data set provided on OSF suggests that the results may indeed be influenced by a small number of extreme values which have survived data trimming. I elaborate on this issue in considerable detail below, and have included some suggestions for how this issue can be fixed. Another, relatively minor issue which should still be addressed is that, while readability of the manuscript overall has considerably improved, the Results sections for the five experiments are still difficult to follow and contain a multitude of tables and figures.

*** We are grateful to Reviewer 1 for these comments. We changed the trimming of data in both the word-by-word SPR experiments in the revised version of the manuscript (see below in detail). 

In sum, I have read the revised version with great interest, and I believe that the five experiments reported in the manuscript potentially constitute a valuable contribution to the field. That said, the issue of outlier exclusion strikes me as particularly important: It obviously makes a crucial difference whether the garden-path effects reported in the manuscript really exist, or whether they are caused by a somewhat larger proportion of extreme values in the garden-path than in the control conditions. The issue therefore has to be addressed in a responsible manner.

*** Thank you. The more conservative trimming procedure which we used in the revised version of the paper yielded exactly the same pattern of results as the previous liberal trimming (see below in detail). 

I. Outlier Exclusion

- In the first version of the manuscript, the authors had relied on an extremely liberal method for outlier exclusion, and had excluded only very extreme reading times above 15000ms. Even in the revised version, the cut-offs applied (6500ms for Experiment 3, 5500ms for Experiment 5) are far too liberal for web-based self-paced reading experiments of this kind. Given that the critical disambiguating regions in these experiments consist of a single 6-8 letter word, even a reading time of more than 2 seconds is a clear indication that the measure does not reflect natural reading. For the present study, however, even a reading time of 5000 ms is not excluded from the analysis. The authors justify their approach to outlier exclusion by arguing that they have followed the recommendations from Baayen & Milin’s (2010) tutorial for reaction time analysis. However, applying the approach suggested in this tutorial to the present data set mechanically (i.e. without considering what the measures actually represent in this specific study) strikes me as not quite suitable, for at least two reasons: First, Baayen & Milin’s recommendation to only conduct minimal data trimming and to remove only clearly discontinuous reading times refers to the analysis of reaction times in general, and does not take into account what these reaction time measures represent in a concrete study (such as properties of the stimulus for which the reaction times were measured). Thus, their recommended approach is based exclusively on the distribution of data points, and does not discuss any content-based common-sense criteria for what constitutes an outlier. For the reading times measured in the present study, we know that a healthy, fully literate, adult native speaker of a language should definitely be able to read a single 7-letter word in considerably less than 2 seconds. As a result, an extreme reading time of, for instance, 5000ms should definitely be considered an outlier, irrespective of the distribution of data points. Second, Baayen & Milin’s tutorial refers to experimental reaction time studies conducted in a controlled lab environment. Data sets from such studies typically contain only a relatively small number of outliers because the controlled surroundings ensure that a participant only rarely gets distracted from the task. In the current web-based study, in contrast, the experimenter had no control of the participant’s surroundings. As a result, outliers are a potentially much more severe issue in web-based studies. In sum, I thus strongly recommend that the authors revise their analyses and rely on an established approach to outlier exclusion used in previous self-paced reading studies (e.g. first exclude extreme values which clearly do not reflect natural reading, then also exclude reading times which are more than 2 standard deviations above or below the overall mean reading time for the respective segment).

Note that the influence of outliers is a particularly crucial issue here, particularly given that linear-mixed effects models are known to be fairly vulnerable when it comes to the influence of a small number of extreme data points. I thus conducted a brief analysis for the critical disambiguating segment in Experiment 3 myself, with the data provided by the authors on OSF. The results from this more conservative analysis suggest that the crucial effect of ambiguity reported for the disambiguating segment in Experiment 3 is at least partly driven by the influence of a small number of outliers in the data, and becomes considerably smaller when these outliers are excluded: If I conduct the analysis in the same way as in the manuscript (i.e. exclusion of data points above 6500 ms and below 150 ms), mean reading times by condition for the disambiguating segment (i.e. Segment 7) obviously show the very same data pattern as in Figure 2:

Garden-path, animate: 694 ms (SD: 642)

Non-garden-path, animate: 602 ms (SD: 462)

Garden-path, inanimate: 625 ms (SD: 521)

Non-garden-path, inanimate: 548 ms (SD: 376)

These results show clear numerical trends for a garden-path effect, with a 92 ms difference between garden-path- and non-garden path trials for animate items and 77 ms for inanimate items (Note, however, that the huge standard deviations already suggest a presence of a small number of extreme data points.). However, if data points which are more than 2 standard deviations above or below the overall mean reading time for the segment are excluded (an established procedure for outlier exclusion used in a number of previous reading studies), the respective garden path-effects are very considerably smaller, with only a 27 ms difference for animate and 25 ms for inanimate items:

Garden-path, animate: 558 ms (SD: 303)

Non-garden-path, animate: 531 ms (SD: 272)

Garden-path, inanimate: 534 ms (SD: 275)

Non-garden-path, inanimate: 509 ms (SD: 234)

In sum, all this calls for a through exploration of the potential influence of outliers in the data, for all experiments reported in the manuscript. It is crucial to determine whether the key effects reported in the manuscript really reflect longer average reading times for garden-path- than for non-garden-path sentences, or whether the effect is instead driven by a slightly higher number of outliers in the garden-path conditions.

*** We would like to thank the reviewer for such a thorough comment. We fully agree that the selected data trimming procedure might affect the measured effects and that it therefore is a delicate issue. Initially, our approach aimed to exclude only unmistakable outliers. However, we acknowledge the reviewer's point that this method may inadvertently include extreme values resulting from external factors like distraction, rather than solely reflecting processing costs.

In light of the reviewer's recommendations, we proceeded with the following revisions: Firstly, we excluded all response times (RTs) below 150 ms. Secondly, to approximate a normal distribution, we performed a log transformation on the RTs. Thirdly, we determined the upper trimming value as the mean of log-transformed RTs plus 2.5 standard deviations. For Experiment 3, this value corresponded to 1486 ms, while for Experiment 4, it was 1600.2 ms. Subsequently, we reran the models and updated the files on OSF. Importantly, the pattern of results remained entirely consistent, with independent effects of sentence type and object animacy once again observed.

Additionally, we have also revised the data trimming procedure for Experiments 2 and 4, where sentence-as-a-whole presentation was employed, so that it now aligns with the 2.5 * SD criterion. 

Additional minor issues:

I.

- In their response letter, the authors say that, based on reviewer suggestions in the previous review round, they have revised Figures 2 and 5 so that they show only segments after the experimental manipulation, and also display untransformed reading times instead of inverse-transformed values. However, these updated graphs are not included in the revised version of the manuscript. I suspect this is just due to some sort of converting error.

*** Thanks for noticing this! It was our mistake since we forgot to change these figures while submitting the revision. This is now corrected.

II.

- The analyses are still considerably too detailed; the revised manuscript contains far too many tables and graphs. Some of the tables should either be merged, included in the supplementary materials, moved to OSF, or deleted entirely (with the key results instead reported in the text).

*** We removed the full models from the Supporting information, since these may be easily reproduced from the data available on OSF. Moreover, Tables 2, 5, and 7 (containing acceptability judgment scores and RTs in Experiments 2 and 4) were merged together. And we also merged Tables 10 and 12 (containing error types in response to qwhat in Experiments 4 and 5). This led to a substantial reduction of the number of tables in the manuscript. 

III.

- As suggested in the previous review round, the authors have added an additional motivation for re-doing Experiments 2 and 4 with self-paced reading (i.e. Experiments 3 and 5) on page 19 (ll. 752-765). While it strikes me as reasonable to motivate these studies with reference to the possibility to look at garden-path effects separately for each segment, arguing that self-paced reading is interesting because it rules out the possibility to go back to a previous segment of the sentence feels a bit weird to me. This essentially characterizes self-paced reading as an artificial research method which does not allow participants to read the sentences naturally.

*** We agree that this was not formulated ideally. We now revised and (based on the suggestion of Reviewer 2 also shortened) this part of the text. The current version is: 

“Experiment 3 was a replication of Experiment 2 with one important difference. The stimuli were not presented in a sentence-at-once mode, but using a classic word-by-word presentation. Such presentation provide us with important information about the course of processing, which will be crucial for the interpretation of our results. Measuring RTs on specific regions of the sentence can also give us information about additional difficulty in processing that relates to the changes of animacy of the disambiguating noun.”

IV.

- In the paragraph discussing issues of statistical power (particularly the number of items per condition each participant encounters during test session) in the Discussion section on page 39, it strikes me as unnecessary to characterize this as a crucial limitation. My suggestion would be to instead include a brief explanation of why it was not possible or problematic to construct a larger set of materials, for instance to do properties of the language.

*** Thanks for this suggestion. We would prefer to leave the part on task adaptation as it is, but we added this text to the last paragraph of General discussion:

“Moreover, the creation of experimental items was already highly constrained due to grammatical factors. As we mention in the section Structure of Czech and the Current Study, the number of verbs that can be plausibly combined with both animate and inanimate patients is limited in Czech, and thus, we had to use each verb and disambiguating noun twice to get the current number of experimental items. To get even more items and thus raise the statistical power, we would also need to repeat the ambiguous nouns since they come from a feminine declension class which contains a relatively closed set of nouns denoting people (i.e., the růže-paradigm).”

Reviewer #2:

The authors have carefully revised the manuscript and addressed my concerns very well. I think that the manuscript has much improved, particularly in the interpretation of the evidence. I only have remaining minor comments and suggestions.

*** We would like to thank Reviewer 2 for their valuable comments and suggestions.

Minor points

I.

1) The manuscript has become quite lengthy. Of course, it is up to the editors to decide if that is fine, but my advice is to shorten it to improve readability. These are some examples of places that may be shortened:

- Detailed descriptions of previous studies in the Introduction;

- Motivation of design of Experiment 3 (lines 752-765);

- Perhaps some more of the overlapping information in Methods of Experiments 3-5;

- Discussion of methodological aspects in the Discussion section (lines 1421-1497) – these are important but can be expressed with fewer words.

*** We reduced the motivation of design of the Experiment 3 and we partly reduced the Hypotheses section of Experiment 5 (by referring to the Hypotheses of Experiment 3). Also, we removed certain parts of the Introduction. However, we would prefer to leave the General Discussion as it is.

II.

2) I would like to clarify my minor point #2 from the previous review round. I meant not only individual differences between readers in whether they are susceptible to garden-path effects but also across-trial differences in whether the reader follows the garden path. In other words, I meant that possibly in some trials, the participants did not follow the garden path at all and parsed the sentence correctly in the first place. Perhaps the authors could consider whether any of the findings could be accounted by this processing pattern, rather than by any patterns of recovery from being garden-pathed. However, I do not insist on incorporating this into the manuscript.

*** We agree with the reviewer that this question is interesting. In part, we touch it in the last paragraph of the General discussion where we mention the task adaptation effects. Nevertheless, we believe that our data at hand do not enable us to discuss these issues in depth and since the paper is already too long (as both reviewers point out), we would rather leave it out of our scope. 

III.

3) Phrasing of some of the hypotheses has become somewhat vague:

- Hypothesis of Experiment 1 (line 573-574): “there will not be a big difference any significant differences in response accuracy”;

- Hypothesis of Experiment 4 (line 945): “there should be no (or little) difference in response accuracy”.

I feel that it is quite vague to discuss “big differences” or “little differences” (unless we are discussing effect sizes, which is not the case here), so wording in terms of statistical significance would be more rigorous.

*** We agree. In the revised version, it was changed accordingly.

IV.

4) I apologize for missing this issue at my first reading but I do not agree with the authors’ description of heuristics with reference to Ferreira & Patson (2007). The authors claim that the heuristic is “the tendency to not inhibit the initial misanalysis of garden-path sentences”. But Ferreira & Parson (2007) use the term ‘heuristics’ to describe the parsing mechanisms per se (for example, late closure heuristic, minimal attachment heuristic, reliance on plausibility, etc.) rather than any mechanisms used to deal with conflicting parses. In other words, the initial misanalysis of garden-path sentences is indeed created based on heuristics but the tendency to not inhibit the initial misanalysis is a separate phenomenon.

*** We believe that this is probably a misunderstanding. We do not claim that heuristics is “the tendency to not inhibit the initial misanalysis of garden-path sentences”. In the last version of the paper, there was “It has been pointed out multiple times that one such example of heuristic processing is the tendency to not inhibit the initial misanalysis of garden-path sentences.” which we think is in accordance what Ferreira & Patson (2007) claim. Nevertheless, to avoid such a confusion and to be more precise, we changed “example of heuristic processing” to “outcome of heuristic processing” in the revised version of the manuscript.

V.

5) I suggest some editorial changes:

- Lines 1084-1085: “These results are once 1084 again in line with previously mentioned accounts of garden-path sentence processing” – Please clarify which accounts are meant.

*** We added several citations to this sentence.

- Line 1322-1323: “Slattery and colleagues [21] do not claim that reanalysis of GP sentences does not fail occasionally” – Double negation is difficult to process, please rephrase.

*** Changed to “Slattery and colleagues [21] do not claim that reanalysis of GP sentences is always successful.”

- Lines 1337-1339: “It has already been shown that comprehenders differ tremendously in what (if any) re-reading strategies they use during reanalysis of GP sentences [42,43]” – This sentence looks stranded in this particular spot. I would consider moving this idea to the new paragraph on individual differences (line 1410 and further).

*** Thanks for the suggestion. We moved the sentence to the paragraph about individual differences.

VI.

6) Please proofread the paper for punctuation and possibly typos, for example:

- Line 270: “open ended” – A hyphen is missing.

*** Hyphen added.

- Lines 452-454: “as well as the experimental items being created specifically for the purposes of our experiments and did not come from actual usage” – Perhaps “were created” is meant.

*** The paragraph was revised. The current version is:

“As Experiment 1, we ran a pilot study using the acceptability judgment task. Since the structure we worked with has not, to our knowledge, been used in any previous work, and the experimental items were created specifically for the purposes of our experiments, we needed to ensure that the unambiguous versions of the sentences will be considered unproblematic by the readers.”

- Line 1086: “However, we would like to note, that we also detected” – There is an unnecessary comma.

*** Comma removed.

- Lines 1214-1215: “for example while answering the comprehension questions” – A comma is missing.

*** Comma added.

- Lines 1274-1275: “(they successfully attached the 1274 disambiguating noun to the structure and they also reanalyzed the ambiguous noun.)” – The full stop should follow the bracket.

*** Corrected.

- Line 1346-1347: Perhaps there should be no paragraph break here.

*** Agreed. The two paragraphs are now merged.

- Line 1371: “similarity based interference” - A hyphen is missing.

*** Corrected.

---

## [Editor Report · Decision Letter 2]

5 Jul 2023

Garden-path sentences and the diversity of their (mis)representations

PONE-D-23-03278R2

Dear Dr. Ceháková,

We’re pleased to inform you that your manuscript has been judged scientifically suitable for publication and will be formally accepted for publication once it meets all outstanding technical requirements.

Kind regards,

Claudia Felser, Ph.D

Academic Editor

PLOS ONE
---

## [Editor Report · Acceptance letter]

7 Jul 2023

PONE-D-23-03278R2 

Garden-path sentences and the diversity of their (mis)representations 

Dear Dr. Ceháková:

I'm pleased to inform you that your manuscript has been deemed suitable for publication in PLOS ONE. Congratulations! Your manuscript is now with our production department. 

Kind regards, 

on behalf of

Dr. Claudia Felser 

Academic Editor

PLOS ONE